# Hydrogen sulfide as a potent predator-derived kairomone mediating fear-related responses in mice

Ana Catarina Lopes, Julien Brechbühl , Aurélie de Vallière, Noah Gilliand, Flavio Ferreira &
Marie-Christine Broillet ✉

Olfaction plays a critical role in survival across species, notably in threat detection. Volatile olfactory molecules signaling the presence of a danger in the environment share a specific chemical signature, particularly sulfur-containing moieties detected by the mouse olfactory Grueneberg ganglion (GG) neurons. Our study focuses on one of the most toxic air pollutant, hydrogen sulfide ($H_2S$). We reveal here a novel facet of $H_2S$ which acts as a danger signal, a kairomone alerting the prey for the presence of nearby predators as $H_2S$ can originate from meat-eater secretions. $H_2S$ activates the cyclic nucleotide-gated channels (CNGA3) present on the sensory cilia of GG neurons. This direct channel opening lets calcium into the cells ensuring neuronal activation and signal transmission to specific brain regions associated with stress and fear-related behaviors. Moreover, using GG-axotomized mice, we demonstrate the biological relevance of GG neurons to detect $H_2S$. These results give new insights into predator-prey dynamics and danger communication which is fundamental for the survival of the species.

Hydrogen sulfide ($H_2S$) is, with nitrogen dioxide ($NO_2$) and sulfur dioxide ($SO_2$), one of the most toxic air pollutants which causes multiple dangerous effects at very low concentrations[1]. $H_2S$ is produced naturally by environmental sources (Fig. 1a) such as the eruptions of volcanoes or the geothermal activity. Chronic low exposure to volcanic $H_2S$ has been implicated in cardiovascular diseases[2]. Production of environmental $H_2S$ can also happen in sulfate-reducing bacteria through anaerobic digestion of organic material[3]. These environmental origins represent the most important part of $H_2S$ emission. It is also known that $H_2S$ production can occur endogenously by the intestinal microbiome (Fig. 1a). Indeed, sulfate-reducing bacteria such as *Escherichia coli* naturally present in our microbiome produce $H_2S$ implicated in many physiological processes[4]. At high concentrations, endogenous $H_2S$ is involved in different gastrointestinal diseases[5].

H₂S is characterized by a very specific odor of rotten eggs. This distinctive warning odor is recognizable even at very low concentrations. Indeed, this noxious gas is recognized for its extreme sensitivity in olfactory detection across the animal kingdom[6]. In mice, for example, recent studies have shown that it activates a specific class of sensory neurons, the type B cells in the main olfactory epithelium (MOE)[7]. The presence at 0.5 parts per billion (ppb) of $H_2S$ suffices for its detection, while concentrations exceeding 10 parts per million (ppm) induce a cascade of physiological effects, from a

suspended animation-like state at 80 ppm to hypothermia, ultimately leading to death[8].

Interestingly, volatile olfactory molecules signaling the presence of a danger in the environment share a specific chemical signature with heterocyclic sulfur- or nitrogen moieties[9]. These danger signals are involved at two distinctive levels: in intraspecies signaling through alarm pheromones[10],[11] and in interspecies signaling via predator kairomones[12]. Indeed, kairomones are present in the secretions of predators such as in the urines, anal gland secretions or in the feces of these animals and they will induce aversive behaviors in the preys[13].

The notion of kairomones is defined by a chemical communication between two species which will induce in the receivers an adaptive behavior depending on the message sent for their own benefits[14]. From the prey point of view, this danger molecule detection will be beneficial for its survival.

During our recent investigations, we found that the secretions from the anal gland of the skunk and the urine of the raccoon were two potent sources of kairomones and that they displayed a strong sulfuric odor[15]. Indeed, the meat diet will contribute to the production of sulfur-containing volatile substances in the large intestine that will be excreted and found in the feces or urines of the meat eater[16,17]. Interestingly, the endogenous bacterial production of $H_2S$ is also intricately linked to the predator's meat diet,

Department of Biomedical Sciences, Faculty of Biology and Medicine, University of Lausanne, Lausanne, Switzerland.
✉e-mail: marie-christine.broillet@unil.ch

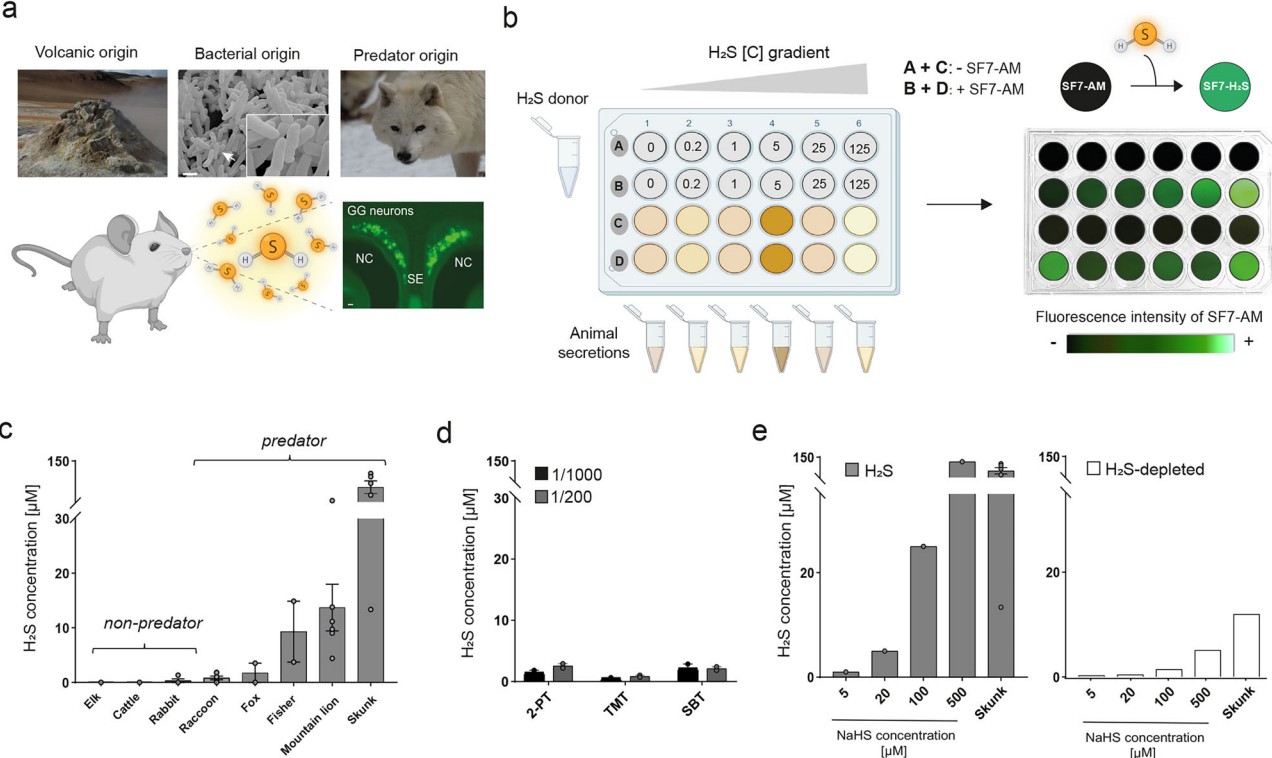

**Fig. 1 | Predator secretions emerge as a potent source of H₂S. a** Volcanic eruptions and the linked geothermic activity are the main environmental sources of $H_2S$ (Volcanic origin). The bacterial production, such as the one from *Escherichia coli (E. coli)* present in the gut microbiota represents the endogenous source of $H_2S$ (SEM micrograph of an *E. coli* bacterial colony; white arrow, localization of the zoomed area; scale bar: 4 µm; Bacterial origin). The potential new source of $H_2S$ released from predator secretions (example: a wolf *Canis lupus*; Predator origin). Schematic illustration of $H_2S$ detection by the mouse (elements of methods created with BioRender). Microscopic image of a coronal section of an OMP-GFP mouse Grueneberg ganglion (GG) where $H_2S$ is detected (NC: Nasal cavity, SE: Septum; scale bar: 20 µm). Acquisitions performed by LED-fluorescence microscopy (EVOS® M5000) with a 10× objective. **b** Original experimental design of $H_2S$ detection in predator secretions. NaHS ($H_2S$ donor) was diluted in water to calibrate the system at different concentrations of $H_2S$ (0, 0.2, 1, 5, 25 and 125 µM) (wells A-B). Pure animal secretions were tested to determine their $H_2S$ concentrations (wells C-D). The $H_2S$ fluorescent chelator, SF7-AM (Sulfidefluor-7 acetoxymethyl ester; 25 µM) was added in the wells B + D. Control without SF7-AM in wells the A + C. Representative acquisitions of the fluorescence of SF7-AM in the presence of $H_2S$ in the ELISA plate wells were obtained with the fluorescence stereomicroscope (Leica M165 FC) at 488 nm. These acquired micrographs have been added to a schematic representation of the plastic plate. Plates images created with BioRender. **c** $H_2S$ concentration in the different animal secretions as means ± SEM (*n* = 1–6 plates) and grouped in two categories: the non-predator urines (Elk (*Cervus canadensis*); Cattle (*Bos taurus*) and Rabbit (*Oryctolagus cuniculus*)) and predator secretions (Raccoon (*Procyon lotor*); Fox (*Vulpes vulpes*); Fisher (*Pekania pennanti*); Mountain lion (*Puma concolor*) and Skunk (*Mephitis mephitis*)). **d** Control of the SF7-AM specificity with 2-PT (2-propylthiethane), TMT (trimethylthiazoline) and SBT (2-*sec*-butyl-4,5-dihydrothiazole) at concentrations of 1/1000 and 1/200 (*n* = 2 plates for each condition). **e** On the left, $H_2S$ concentrations present in standard solutions (NaHS) used as scale reference and in the skunk secretions (Skunk). On the right, $H_2S$ concentrations in the same solutions and in the skunk secretions (Skunk) obtained after degassing $H_2S$ ($H_2S$-depleted).

hinting at a shared origin of production between the alerting molecules and $H_2S$[18,19]. We thus hypothesized here that predator secretions might contain high levels of $H_2S$, which could play the role of a potent kairomone allowing predator avoidance (Fig. 1a).

The previously identified alerting molecules such as the alarm pheromone 2-*sec*-butyl-4,5-dihydrothiazole (SBT)[9] and the kairomones trimethylthiazoline (TMT) from the fox[20] as well as 2-propylthiethane (2-PT) from the stoat[21] have their initial detection performed by the neurons of the Grueneberg ganglion (GG; Fig. 1a), positioned at the nasal tip[22]. The GG plays a pivotal role in detecting specific volatile cues associated with danger, especially those containing sulfur motifs[23]. Positioned at the forefront of the nasal cavity, this specialized sensory organ serves as an initial sentinel, equipped to recognize, and process chemical signals indicative of potential threats[24]. Its neurons, nestled within a delicate epithelial layer, possess unique morphological features, including non-motile, invaginated cilia, distinctly different from those present in the MOE[25].

Here, we demonstrate that predator secretions are a source of $H_2S$. We also observed that this gas is detected by specific olfactory neurons, the GG neurons. It opens their sensory transduction channels, the CNGA3 (cyclic nucleotide-gated channels A3). Further activation of the GG-associated brain regions induced fear-related behaviors in mice. Moreover, using GG-axotomized mice, we demonstrate the biological relevance of GG neurons to detect $H_2S$. The detection of this gas, a new kairomone, might be crucial for the prey survival in the wild and for its adaptive behavioral responses.

## Results

### H₂S is present in the body secretions of predators

We have established a method to detect and quantify $H_2S$ in different body secretions from animals using spectrofluorimetry with a specific $H_2S$ probe, sulfidefluor-7 acetoxymethyl ester (SF7-AM)[26]. Different concentrations of the $H_2S$ donor, NaHS[27], were used to calibrate the system (Fig. 1b). We tested different animal secretions coming from either non-predators: elk (*Cervus canadensis*), cattle (*Bos taurus*) and rabbit (*Oryctolagus cuniculus*) and from predators: raccoon (*Procyon lotor*), fox (*Vulpes vulpes*), fisher (*Pekania pennanti*), mountain lion (*Puma concolor*) and skunk (*Mephitis mephitis*) of mice (Fig. 1b). We used urines as the animal secretions except for the skunk, where anal gland scents were used. Predators and non-predators were selected according to their diet, meat or non-meat-eating habits.

The biological secretions from predators, most particularly from the mountain lion and the skunk contain a high concentration of $H_2S$ (15 and 75 µM respectively; Fig. 1c). Interestingly, the skunk is known to use the

strong sulfuric odor emitted by its spray as a mechanism of defense[28]. As expected, we did not find $H_2S$ in non-predator secretions (Fig. 1c) as it is mostly produced by meat degradation[17]. The urine and the secretions of predators contain numerous alerting cues with sulfuric molecules in their chemical composition[9,29], we thus confirmed the specificity of SF7-AM and found no interaction with the two kairomones 2-PT and TMT as well as no interaction with the known alarm pheromone, SBT, who shares a similar chemical structure as the kairomones[9] (Fig. 1d). SF7-AM detects $H_2S$ in its gaseous form and degassing the NaHS solutions ($H_2S$-depleted) or degassing the secretions from the skunk led to the observation of a very large decrease in $H_2S$ quantification ($-85\%$ of $H_2S$ in the degassed skunk secretion; Fig. 1e).

We thus identified the presence of $H_2S$ in predator secretions, a new origin of production of $H_2S$. In the wild, the release of this gas might be detected by the GG neurons of the preys to warn them from a nearby danger (Fig. 1a).

## $H_2S$ activates the mouse GG neurons

To assess the potential olfactory detection of $H_2S$ by the GG neurons, we stimulated in vivo the mice and then performed immunostainings on GG tissue slices using the phosphorylation of the ribosomal protein S6 (rpS6), as a marker of neuronal activity[30,31]. We used tissue slices of OMP-GFP mice[32] where the green fluorescent protein (GFP) reporter is under the control of the olfactory marker protein (OMP) promoter, which allows for the visualization of mature olfactory sensory GG neurons in green (Fig. 2a). We observed a significant increase of the neuronal activity of GG neurons when mice were stimulated with $H_2S$ compared to the $H_2O$ (control) stimulation (Fig. 2a).

The neuronal activity induced by $H_2S$ was also observed in ex vivo calcium imaging experiments when we perfused different concentrations of the gas on GG tissue slices from OMP-GFP mice (Fig. 2b). We found that $H_2S$ induced significant increases in the intracellular calcium of GG neurons dependent on its concentrations (Fig. 2b). These results showed that $H_2S$ is detected by GG neurons reinforcing the idea that $H_2S$ acts as an alerting molecule, mimicking the kairomone effect of, for example, TMT (Fig. 2b).

We then stimulated GG neurons with $H_2S$, with the alarm pheromone SBT and with the kairomones, 2-PT and TMT. It was already demonstrated that responses induced by TMT were dependent of the presence of extracellular calcium[33]. Here, we showed that all responses to alerting cues tested, including $H_2S$, were reduced in the absence of external calcium as we could see in a representative neuron showing the calcium transients observed (Fig. 2c). Therefore, extracellular $Ca^{2+}$ appears to be important for GG neuron activation after $H_2S$ perfusion, as well as for the other danger cues, as the amplitudes of the responses are significantly reduced (amplitude decreases of alerting cues: SBT, $-65\%$; 2-PT, $-100\%$; TMT, $-45\%$ and $H_2S$, $-24\%$; Fig. 2d). These results suggest that the signaling pathway involved in $H_2S$ recognition needs the activation of an effector allowing $Ca^{2+}$ entry.

We next investigated if a $H_2S$-depleted solution could still activate GG neurons and observed that intracellular calcium entry in GG neurons was significantly reduced compared to $H_2S$ (amplitude decrease: $-75\%$; Fig. 2e).

The secretions of the skunk contained the highest concentration of $H_2S$ (Fig. 1c), and we demonstrated earlier that they induced an increase in the intracellular calcium of GG neurons[15]. We then found that the contribution of $H_2S$ to this GG neuronal activation was significant by depleting $H_2S$ from the skunk secretions. Indeed, we found an amplitude decrease of 50% in the observed calcium transients compared to the responses induced by native skunk secretions (Fig. 2f).

These results demonstrated that $H_2S$ activates GG neurons. We then tried to identify the molecular target of $H_2S$ in these neurons.

## $H_2S$ activates the cyclic nucleotide-gated channel A3 (CNGA3) via S-sulfhydration

A series of ion channels and neuronal receptors are key target proteins for S-sulfhydration by $H_2S$ leading to their activation or inhibition and subsequent modification of neuronal functions[34,35]. According to a recent study in the olfactory system, the cyclic nucleotide-gated channel A2 (CNGA2) is important for $H_2S$ detection in the MOE[7]. We thus decided to test the direct effect of $H_2S$ on the CNG channel present in the sensory cilia of GG neurons, the CNGA3[36] (Supplementary Fig. 1). To achieve that, we used a heterologous system with Human Embryonic Kidney cells (HEK 293) in which we transfected a reporter plasmid encoding for GFP and a second plasmid coding for the murine CNGA3. After 24–48 h of transfection, the cells were used for calcium imaging experiments (Fig. 3a). We performed immunocytochemistry on transfected HEK cells and confirmed the membrane expression of CNGA3 (Fig. 3b). We also tested the functionality of the expressed CNGA3 perfusing 8-bromo-cGMP, a cyclic nucleotide which is membrane permeant, and we observed an increase in the intracellular calcium only in CNGA3-positive cells (Fig. 3c).

We then perfused different concentrations of $H_2S$ and observed an increase in the intracellular calcium levels of CNGA3 expressing cells (Fig. 3d). Considering the number of responding cells and the amplitude of the responses to $H_2S$, we observed a dose-dependent activation (Fig. 3e). To ensure that the responses observed with $H_2S$ are due to a direct activation of CNGA3 and not to an indirect increase in cyclic nucleotides by a potential inhibition of phosphodiesterase (PDE)[7], we used perfusions of an IBMX gradient (from 2 μM to 200 μM) on CNGA3-postive cells. In the presence of this PDE inhibitor, we did not observe any increase in the intracellular calcium up to 100 μM of IBMX, a concentration known to efficiently inhibit PDE[7] (Supplementary Fig. 2a, b). We then tested the activation of the CNGA3 channel by $H_2S$ in the presence of IBMX, at this 100 μM concentration and observed no significant differences in the responses with or without the inhibitor (Supplementary Fig. 2c, d), confirming that the observed $H_2S$-related responses were independent of a PDE activity. Moreover, other known GG ligands such as 2-PT, SBT and TMT were not able to directly activate CNGA3 (Fig. 3f). Taken together, our observations strongly suggest that $H_2S$ acts as a direct activator of the CNGA3 present in the cilia of mouse GG neurons.

To determine if $H_2S$ activates CNGA3 by S-sulfhydration, we next established a colorimetric assay using 5,5'-dithio-bis-(2-nitrobenzoic acid) (DTNB)[37], a molecule reacting with free sulfhydryl groups, in this case, free cysteine residues of CNGA3. In this assay, the reaction of DTNB with SH- groups leads to the production of mixed disulfide and 2-nitro-5-thiobenzoic acid (TNB), a yellow-colored product with an absorbance at 412 nm[38] (Fig. 3g; left panel). We used the lysate from CNGA3-transfected cells and added DTNB (0.3 mM). When DTNB binds to CNGA3 cysteine residues, we observed an increase in the relative absorbance over time, meaning that CNGA3 possesses free SH-groups (Fig. 3g; right panel). As negative control for our assay, we tested the competition between DTNB and iodoacetamide (IAA), which binds irreversibly to SH- groups[39] and we did not observe any relative absorbance (Fig. 3g). Adding $H_2S$ as a competitor of DTNB decreased the production of TNB as we observed a significant decrease in the relative absorbance (Mann–Whitney two-tailed test; *p = 0.0022*) (Fig. 3g). This suggests that $H_2S$ binds to SH- groups of CNGA3 in a reversible manner. Relative absorbance of the negative control was significantly different compared to DTNB condition and DTNB + $H_2S$ condition (Mann–Whitney two-tailed test; $p = 0.0022$).

These results show that $H_2S$ directly opens the calcium channel CNGA3 inducing cellular activation. A similar process could take place in GG neurons. We then investigated the integration of this $H_2S$ danger signal in the GG-associated brain regions involved in physiological fear responses and fear-related behaviors.

## $H_2S$ activates the brain areas involved in the circuitry of chemical danger detection

We focused on three brain regions (Fig. 4a). The first region of interest was the olfactory bulb (OB). The GG neurons project their axons to specific clusters within the OB, forming synapses in discrete regions known as necklace glomeruli (NG; Fig. 4b)[40,41]. These NG exhibit a distinct anatomical

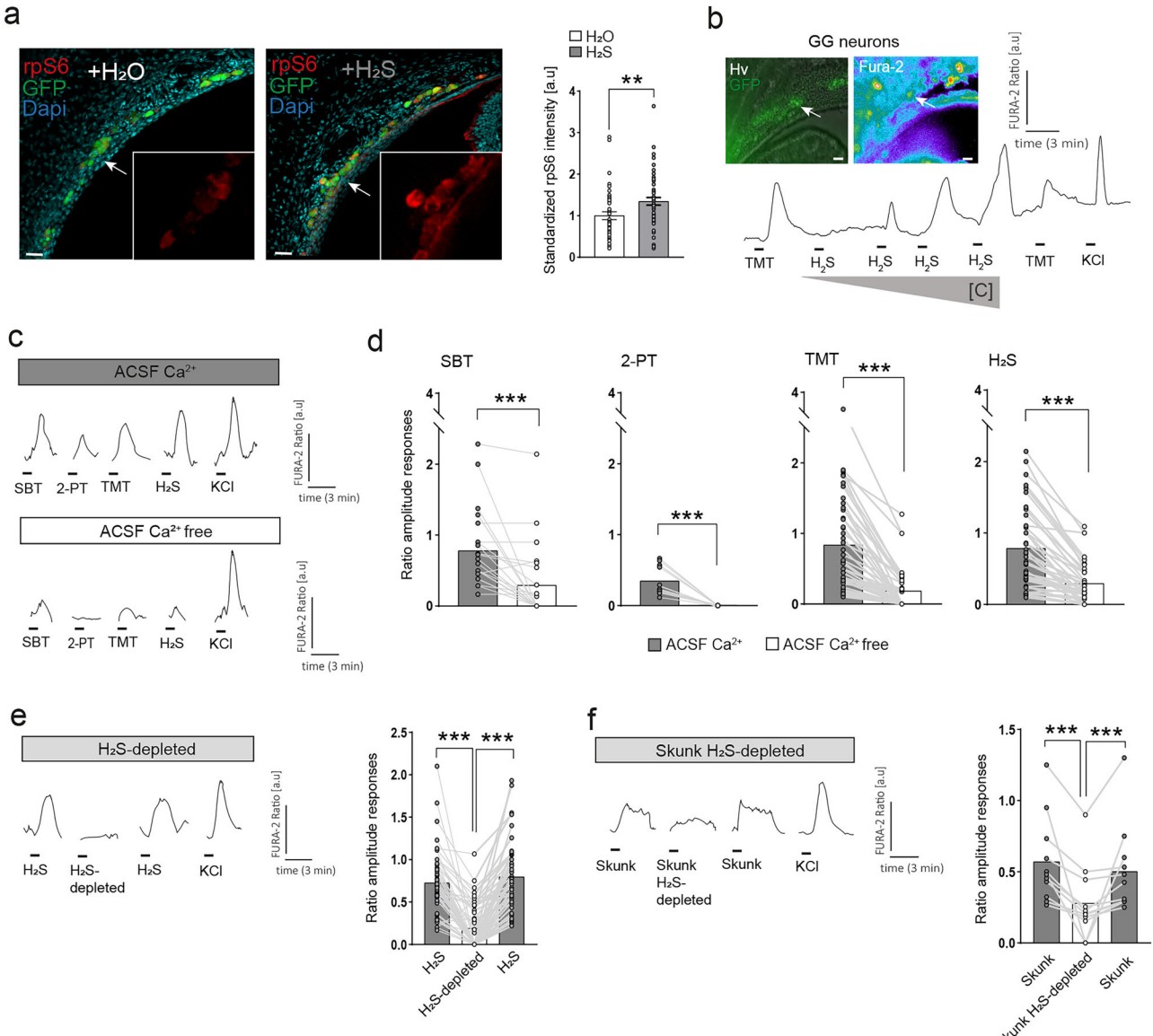

**Fig. 2 | H₂S is detected by the mouse GG neurons. a** Immunostainings on GG tissue slices from OMP-GFP mice to test the neuronal activity using the rpS6 Marker after stimulation with H₂O and H₂S. Green, GFP expression in mature olfactory neurons. Blue, Dapi staining. White arrows, localization of zoomed areas. Scale bars: 20 μm. Quantification of rpS6 intensity for each neuron after H₂O (control in white; $n = 46$ neurons; number of pups = 3) and H₂S (grey; $n = 53$ neurons; number of pups = 3) stimulation. Values represented as mean standardized with aligned dot plots ±SEM. A two-tailed Mann–Whitney test was used to compare the two stimulations: **$p < 0.01$. **b** Microscopy images of a tissue slice of a mouse GG observed with a DIC contrast (Hv) showing neurons in green (GFP) and after Fura-2 loading. Scale bars: 20 μm. Intracellular calcium changes observed in one representative GG neuron (white arrow) with a TMT perfusion (100 μM), different H₂S concentrations (1, 5, 12.5 and 25 μM) and a KCl perfusion (10 mM) to verify neuronal viability. Fura-2AM ratio (arbitrary units) and time (minutes). Bars indicate the perfusion times. **c** Intracellular calcium changes observed in one representative GG neuron with SBT,

2-PT, TMT (1/5000), H₂S (25 μM) or KCl (10 mM) in a ACSF Ca²⁺ *vs.* in a ACSF Ca²⁺ free solution. **d** Ratio amplitude responses for each tested substance with in a ACSF Ca²⁺ *vs.* in a ACSF Ca²⁺ free solution for each neuron (SBT, $n = 23$; 2-PT, $n = 6$; TMT, $n = 45$; H₂S, $n = 42$ neurons; number of pups = 3). Grey light lines connect the same neuron for the different tested conditions. **e, f** Intracellular calcium changes observed in one representative GG neuron after degassing the solution of H₂S (25 μM) (H₂S-depleted; **e**) and degassing the skunk secretions (Skunk H₂S-depleted solution; **f**). Bars indicate the perfusion times. Ratio amplitude responses for each GG neuron for H₂S (25 μM), H₂S-depleted and H₂S (25 μM) (**e**; $n = 51$; number of pups = 3) and for the experiments with the secretions of the skunk, skunk H₂S-depleted and the skunk again (**f**; $n = 11$; number of pups = 1). Grey light lines connect the same neuron for the different tested conditions. Values represented as mean values with aligned dot plots in a before-after graph; a paired Wilcoxon test used, ***$p < 0.001$ (**d–f**).

arrangement, appearing as a chain-like pattern encircling the dorsal, medial, lateral and ventral OB[42] and serves as specialized relay points where the information from the GG is processed and integrated into the olfactory circuitry[43]. To observe the neuronal activity in the OB, we used cFOS staining[44] on OB tissue slices after stimulating the mice in vivo with H₂O (control) or H₂S (125 μM). Compared to the baseline activation observed with H₂O stimulation, we observed a significant increase in the number of

cFOS-positive cells (+220% of activation) surrounding glomeruli after stimulation of animals with H₂S (Fig. 4b, d). For this analysis, we considered only the glomeruli that were immunoreactive for phosphodiesterase 1a (Pde1a), a specific marker of the NG[45] (Supplementary Fig. 3).

We next investigated the cerebrum subregions implicated in the processing of danger information[46,47], such as the lateral amygdala (La), the ventromedial hypothalamus dorsomedial nucleus (VMHDM) and the

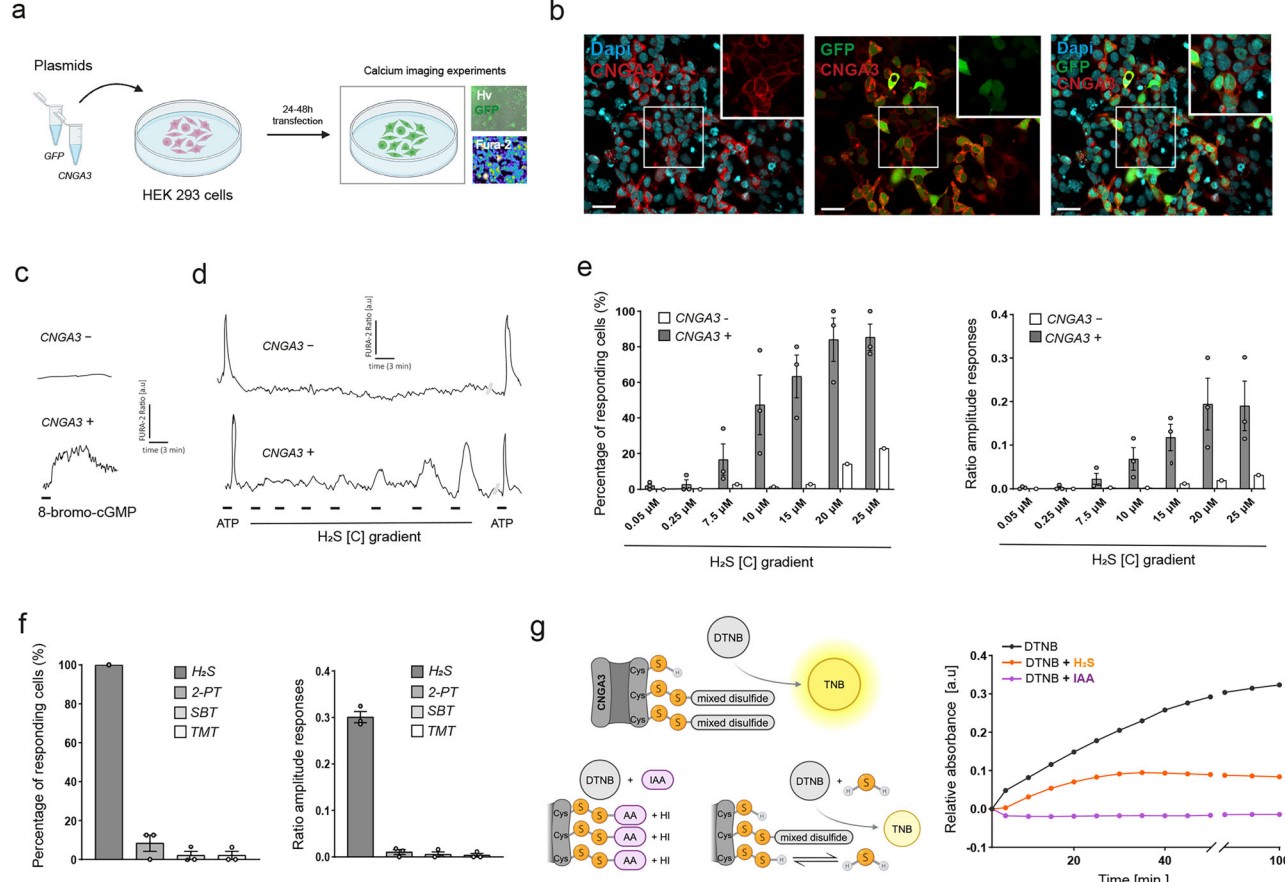

**Fig. 3 | The ion channel CNGA3 is a target of H₂S. a** Transfection of HEK 293 cells with a plasmid encoding for GFP to validate the transfection and a plasmid coding for CNGA3. After 24–48 h, transfected cells are used for calcium imaging experiments (designed by BioRender). Microscopy images of the transfected cells with a DIC contrast (Hv) showing cells in green (GFP reporter plasmid) and after Fura-2 loading. **b** Immunostainings of HEK cells transfected with GFP and CNGA3. White squares: zoomed areas. Green, GFP. Red, CNGA3. Blue, Dapi staining. Scale bars: 20 μm. **c** CNGA3 expressed in HEK cells are activated by 8-bromo-cGMP (1000 μM). Representative examples of the intracellular calcium changes observed in one CNGA3-negative cell (top; CNGA3−) and in one CNGA3-positive cell (bottom; CNGA3+). The Fura-2AM ratio is given in arbitrary units and time in minutes. Bar indicates perfusion time. **d** Representative examples of intracellular calcium changes observed in one CNGA3-negative cell (top; CNGA3−) and in one CNGA3-positive cell (bottom; CNGA3+) in the presence of ATP (100 μM; to control the viability of the cells) and in the presence of different concentrations of H₂S (gradient). The Fura-

2AM ratio is given in arbitrary units (a.u) and time in minutes. Bars indicate perfusion times. **e** Graphs representing the percentage of responding cells (left) and the amplitude of the responses (right) to different H₂S concentrations. Number of cells: 50–150 from 1 to 3 different experiments. White bars, CNGA3-negative cells (CNGA3−); grey bars, CNGA3-positive cells (CNGA3+). **f** Graphs representing the percentage of responding CNGA3-positive cells (left) and the amplitude of the responses observed (right) to H₂S (25 μM), and to the other known GG ligands, SBT, 2-PT and TMT (100 μM) (number of cells: 50). **g** Fluorimetric biochemical assessment of CNGA3 S-sulfhydration by H₂S. Schematization of DTNB effect on the free thiol (SH−) groups performed on a lysate of HEK cells expressing CNGA3 (on the left; BioRender). Measurement of relative absorbance (arbitary units; a.u) of TNB over time in minutes at 412 nm (on the right). Three different conditions were tested : DTNB alone (0.3 mM; black line); DTNB + IAA (0.5 mM; magenta line) and DTNB + H₂S (25 μM; orange line).

periaqueductal gray (PAG). While the La zone is implicated in social interactions and plays an important role in anxiety related-behaviors[48], the two other zones (VMHDM and PAG) are implicated in predator avoidance reactions[49,50] such as freezing and immobility but also in the regulation of hormonal levels such as the ones of corticosterone in mice facing stressful situations[51].

We thus used cFOS in these cerebrum regions to verify the neuronal activation following stimulation with H₂S. We showed differential activation in these brain zones (Fig. 4c). Indeed, quantification of cFOS-positive cells demonstrated a significant activation of the VMHDM (+74% of activation) and of the PAG zone (+252% of activation) but not of the La zone after H₂S stimulation (Fig. 4d).

In summary, we observed that H₂S activates GG neurons projecting then their axons to the NG in the olfactory bulb and then we observed an activation on VMHDM and PAG neurons known to play a role in predator-related fear behaviors[52].

## H₂S induces stereotypical stress-related responses and fear-related behaviors

We further analyzed other stress-related responses such as the level of corticosterone in mice serum after exposure with H₂O (control) or H₂S (125 μM). We observed a significant increase of this hormone in the presence of H₂S (+77% of increase; Fig. 5a). Interestingly, this increase was paired with a significant increase in blood pressure; (+21% of increase for the mean blood pressure; Fig. 5b) and with a significant decrease in the heart rate (−11% of decrease; Fig. 5c), a physiological adaptation previously reported as a kairomone response[23,53,54].

We then exposed mice to H₂S in an open field context with a 3-zone division (safety zone, H₂O; central and danger H₂S zones; Fig. 5d, e). Mice were tracked and we could measure their walking distances, the numbers of entries in the zones and the freezing time. As expected, we found that mice displayed fear-related behaviors, as they walked less and spent more time in the "safety zone". Moreover, during their exploration, mice stayed along the

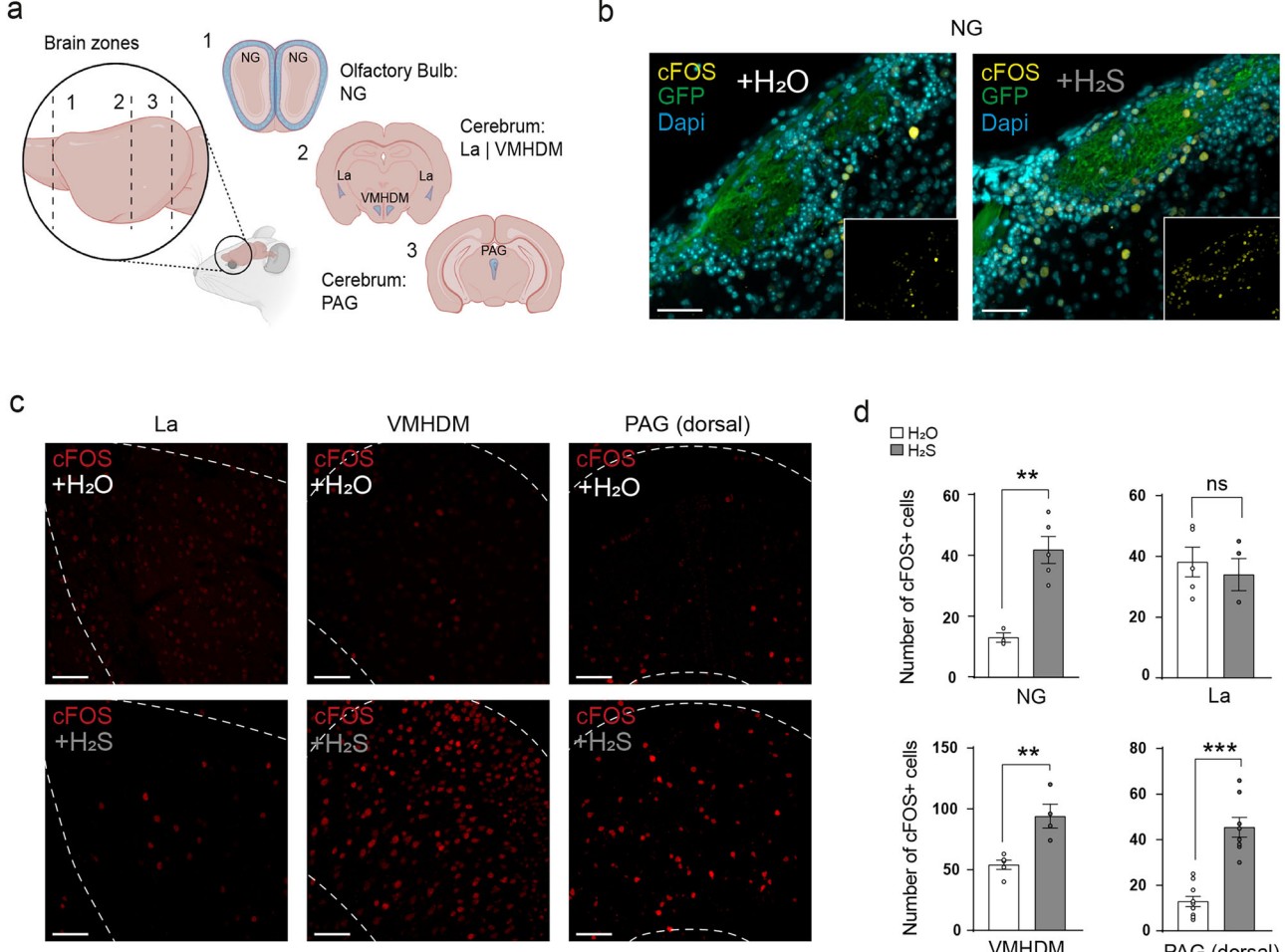

**Fig. 4 | H₂S activates the brain circuitry linked to danger integration.**
**a** Localization of the three brain zones of interest in the mouse. Zone 1, the necklace glomeruli (NG) region of the Olfactory Bulb. Zone 2, Cerebrum with regions such as in the lateral amygdala (La; Bregma: −0.94 to −2.46 mm) and the ventromedial hypothalamus dorsal (VMHDM; Bregma: −1.34 to −1.94 mm) and zone 3, Cerebrum with the periaqueductal gray zone (PAG; Bregma: −2.7 to −3.16 mm). Brain regions by BioRender. **b** Neuronal activation in the NG region after stimulation with H₂O (Control) and H₂S (125 μM). Representative cFOS expression (yellow), axonal processes of olfactory neurons innervating glomeruli in the bulb (green), nuclei stained with DAPI (cyan) showing the neuronal stimulation induced by H₂S; Zoom out to show the general view of cFOS/neuronal activation in zone 1 (**a**). Scale bars:

20 μm. **c** Representative cFOS (in red) immunostainings experiments performed in La, VMHDM (zone 2) and PAG (zone 3) regions after H₂O (control) or H₂S stimulation. Scale bars: 20 μm. **d** Quantification of the number of cFOS-positive cells (cFOS+) observed after H₂O (white bars) and H₂S (grey bars) stimulation for NG ($n = 3$ and $n = 5$ slices for H₂O and H₂S, respectively), La ($n = 5$ and $n = 4$ slices for H₂O and H₂S, respectively), VMHDM ($n = 5$ and $n = 4$ slices for H₂O and H₂S, respectively) and PAG regions ($n = 10$ and $n = 8$ for H₂O and H₂S, respectively). Data are represented as mean ± SEM with aligned dot plots and statistical analysis performed with an unpaired Student's $t$ test two-tailed, ***$p < 0.001$; **$p < 0.01$; ns for non-significant.

borders of the arena avoiding the central zone and they displayed a significant increase in freezing behavior (Fig. 5d, e).

We then verified that the GG was implicated in these fear-related behaviors and performed a nerve axotomy of the GG connections to the OB generating axotomized mice (Axo)[33]. A cookie test first assessed that the MOE-related olfactory abilities were not impacted by this axotomy[55] as Ctrl and Axo mice were equally efficient in finding a buried cookie (Supplementary Fig. 4a).

We then observed these Axo mice in the previously described open field context to verify if these mice lacking GG neurons could still recognize H₂S as a danger cue (Fig. 5f). Interestingly, Axo mice explored the complete arena and had a total absence of fear-related behaviors in the presence of H₂S (Fig. 5f, g and Supplementary Fig. 4b). We finally performed an immunohistochemistry on GG tissues slices from Ctrl and Axo mice to verify the nerve axotomy and therefore confirm the absence of GG neurons in the mice (Fig. 5h and Supplementary Fig. 4c).

In summary, H₂S released by the secretions involuntarily left behind by the predators induces stress responses and fear-related behaviors in mice.

H₂S acts like a kairomone detected by the neurons of the Grueneberg ganglion.

## Discussion

In the wild, mice living in their hostile environment rely on their sense of smell for their vital behaviors such as for finding food, for their social interactions and for finding a mate and most importantly to avoid their predators[13,44]. Sensing danger chemical cues, called kairomones, which are unvoluntary released by these predators in their body secretions (urine, feces), is essential for the survival of the preys[56]. Kairomones are detected by the mouse olfactory system specifically the Grueneberg ganglion which plays the main role in danger detection[24]. Thanks to a combination of spectrofluorimetry for detection, immunostainings showing the neuronal activity and experiments with calcium imaging, we showed here that H₂S is present in predator secretions and that it is detected by GG neurons similarly to the known kairomones 2-PT and TMT.

GG neurons are protected by a keratinized epithelium[11], a gaseous volatile kairomone like H₂S is an ideal ligand to be detected by these

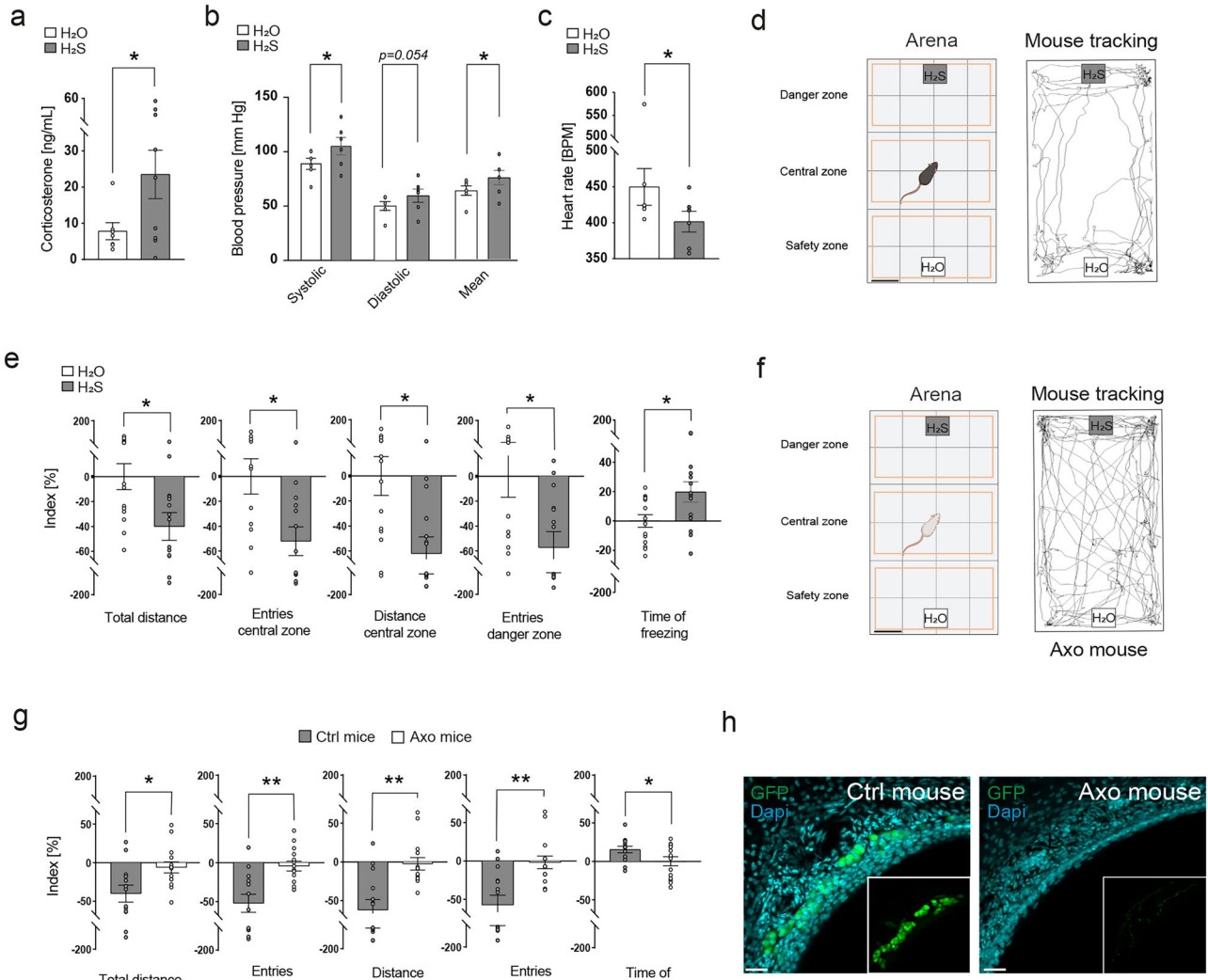

**Fig. 5 | The physiological effects and fear-related behaviors induced by $H_2S$ are dependent of a functional GG. a–c** Effects of the $H_2S$ exposure (125 µM; grey bars) on plasma corticosterone level ($n = 7$ for $H_2O$ and $n = 10$ for $H_2S$) (**a**), on blood pressure (systolic, diastolic and mean) ($n = 6$ for $H_2O$ and $n = 6$ for $H_2S$) (**b**) and heart rate measured in beats per minute (BPM, $n = 6$ for $H_2O$ and $n = 6$ for $H_2S$) (**c**) compared to the exposition to the $H_2O$ (control, white bars). Mean ± SEM with aligned dot plots. A one-tailed Mann–Whitney test (**a**), a paired Student's *t* test (**b**) and a paired Wilcoxon test (**c**) were used to compare the two conditions, \**p* < 0.05. **d** Open field arena (BioRender) delimited in three different zones: a safety, central and danger zone (orange rectangles). Blotting papers with $H_2O$ (white square) or $H_2S$ (grey square) are localized in the middle of the arena width. Scale bar: 5 cm. Mice were tracked for 5 min, and 200 µL of the tested solutions ($H_2O$ or $H_2S$, 25 or 125 µM) were added on the blotting paper. Representative tracking of one mouse (black line) is shown. **e** Quantification of the stress-related behaviors according to

the control and displayed as indexes (%) in the presence of $H_2O$ (in white) and of $H_2S$ (in grey). Total distance travelled, number of entries in central zone, distance travelled in central zone and number of entries in danger zone (125 µM $H_2S$) and total time of freezing (25 µM $H_2S$) were quantified ($n = 14$). Values expressed as mean ± SEM; paired Student's *t* test used, \**p* < 0.05. Axo mice did not show any fear-related behaviors in the presence of $H_2S$ as shown on a representative tracking of one Axo mouse (black line) (**f**) and by the comparison of the different parameters (**g**) between Ctrl mice (results taken from (**e**) for $H_2S$ condition) and Axo mice in the presence of $H_2S$. Values expressed as mean ± SEM; unpaired Student's *t* test or Mann–Whitney test used, \**p* < 0.05; \*\**p* < 0.01. **h** Immunostainings on GG tissue slices to visualize the absence of GG neurons in axotomized (Axo) mice compared to Ctrl mice. GFP, in green, allows the precise localization of mature olfactory GG neurons. In blue, Dapi staining. Scale bars: 50 µm.

neurons. We demonstrated here that $H_2S$ could indeed access and activate GG neurons in vivo as shown by rpS6 staining. We have also shown, by calcium imaging experiments, that this neuronal activation was dose-dependent and reversible. In behavioral experiments, increasing the concentration of $H_2S$ also induced accentuated fear-related responses. These results are in accordance with what is observed in the wild for other fear-inducing cues[57]. Environmental $H_2S$, despite its high volatility, has been shown to have a lifetime that can extend up to 42 days, depending on the weather conditions[58]. We can thus hypothesize that preys will display fear behavior upon each encounter with predator secretion containing $H_2S$.

Once the danger information is recognized by GG neurons localized at the tip of the nose, the $H_2S$ signal will be processed in the OB, more precisely

in the NG innervated by GG neurons. Neuronal activity of the NG was indeed observed when the mice were stimulated with $H_2S$. Moreover, this activation was detectable downstream in other brain regions implicated in stress and freezing behaviors[46,47].

We observed a decrease in heart rate in the presence of environmental gaseous $H_2S$. Interestingly, this result correlates with the different studies done on the impact of perfused $H_2S$ on cardiac functions[59,60] in therapeutic emergency medicine. Moreover, innate fear stimuli such as the exposure to the kairomone 2-methyl-2-thiazoline also have a similar effect on the heart rate[54]. As a physiological response, this heart reaction might participate with the lowering of the body temperature in a life-protective mechanism when the prey cannot flight or fight, leading to a state of suspended-like animation

(death feigning) inducing the lack of interest of the predator[61]. Thus, H$_2$S present in the predator secretions could induce in the prey this artificial hibernation[8,62] as a survival strategy.

H$_2$S is the third identified endogenous gasotransmitter after nitric oxide (NO) and carbon monoxide (CO)[63]. It can be produced by different organs in the body such as the liver or the kidney, from L-Cysteine via three specific enzymes: cystathionine-β-synthase, cystathionine gamma-lyase and 3-mercaptopyruvate sulfurtransferase[64].

Interestingly, at low levels, H$_2$S plays an important endogenous role as a signal molecule involved in many physiological functions and having beneficial effects such as, for example, neuroprotective effects or homeostatic regulatory effects of blood pressure[65]. However, important research, these last few years, showed that this gas, at high endogenous levels, can be both cytotoxic and implicated in several diseases, including neurodegenerative diseases, cardiovascular diseases, and metabolic disorders[66]. Moreover, in the gut, H$_2$S can be produced by the bacteria present in the microbiota[67] and its production is strongly influenced by meat digestion[68]. It is known that excessive production of H$_2$S due to bacterial overgrowth in mammals is involved in colon cancer, ulcerative colitis or Crohn's disease[69]. The H$_2$S produced, associated with intestinal diseases, will then be released, and potentially detected by conspecifics. Thus, this gas could not only play the role of a kairomone which carries a message of danger as it can be found in predator secretions but, further studies, might also show that H$_2$S could have the role of an alarm pheromone, such as SBT, warning the conspecific members of the colony from a potential sick individual. In this context, H$_2$S could, in collaboration with other chemical signals recognized by GG neurons or by other olfactory neurons[70], help detect inflammation and disease in a conspecific.

The activity of multiple ion channels and membrane receptors is influenced by H$_2$S[34,35]. Interestingly, GG neurons express numerous signaling pathways which might contain potential key target proteins[22,33]. We found here that the CNGA3 expressed in GG neurons plays a role in detecting the kairomone H$_2$S as we demonstrated, in a heterologous system where cells expressed CNGA3, that it can be directly activated by H$_2$S. H$_2$S is a gasotransmitter such as nitric oxide (NO) which directly activates the olfactory CNG channels through the modification of sulfhydryl groups[39]. Indeed, it has been demonstrated that exogenous NO upregulates CNGA2 expression and that this is associated with elevated intracellular calcium[71]. Similarly, to this S-nitrosylation reaction, H$_2$S appears to signal through S-sulfhydration as the principal post-translational modification induced[34,35,72]. We demonstrated here, using a colorimetric assay that, H$_2$S binds to free SH- groups of CNGA3 converting them into SSH- groups. We can thus hypothesize that, as the first step of H$_2$S detection, a S-sulfhydration reaction occurs activating CNGA3 in GG neurons. Additional electrophysiological patch-clamp experiments could allow a precise characterization of the ion channel properties of H$_2$S activation and the identification of its S-sulfhydrated cysteine residue(s) after site-directed mutagenesis. As mentioned, H$_2$S has different cardiac functions[59,60] and interestingly, CNGA3 is present in the heart where it could be implicated as a target of the gas[73].

Together, our results identified a new origin of H$_2$S production, the predator secretions. The sensing by the prey of this gaseous cue acting as a kairomone activates specific olfactory neurons, the Grueneberg ganglion neurons. More precisely, the activation of these neurons occurs via the S-sulfhydration of the CNGA3 calcium channels. Further activation of typical brain zones implicated in danger signal integration led to fear-related behaviors and physiological changes. The importance of GG neurons in the detection of H$_2$S was confirmed by experiments performed with axotomized mice. Our study thus sheds light on the important implications of H$_2$S sensing in predator-prey dynamics and on the interplay between olfaction and innate defensive mechanisms.

## Methods
### Spectrofluorimetry
We visualized and quantified H$_2$S in the animal secretions using the chelator Sulfidefluor-7 acetoxymethyl ester[26] (SF7-AM 95%; Sigma-Aldrich). The azide moieties of SF7-AM will react with H$_2$S to generate carboxamide rhodamine 110, which has wavelengths of 488 nm (excitation) and 535 nm (emission). We can then observe the intensity of the reaction depending on the concentration of H$_2$S in the urine or in the presence of NaHS. 100 μl of pure animal secretions collected in the Servion Zoo (Switzerland) and/or ordered online (PredatorPee®, Hermon, ME, USA and Pete Rickard Co., Galeton, PA, USA) were placed in a 96 well plate (Nunc-Immuno™ MicroWell™ solid plate; Sigma-Aldrich, Merck, Germany) and tested with SF7-AM to visualize the fluorescence intensity and determine the concentration of H$_2$S present in each of them. SF7-AM was used at a concentration of 25 μM. As control for the specificity of SF7-AM for H$_2$S, we also tested the chelator on 2-PT (2-propylthietane; Enamine Ltd, Ukraine; stock solution: 10 M), TMT (trimethylthiazoline; Sigma-Aldrich, Merck, Germany; stock solution: 10 M) and SBT (2-sec-butyl-4,5-dihydrothiazole; homemade[9]; stock solution: 10 M) at two different dilutions: 1/200 and 1/1000. Visualization and image acquisitions were performed with a fluorescence stereomicroscope (Leica M165 FC). NaHS (Sodium hydrosulfide hydrate; Sigma-Aldrich; Merck, Germany), a donor of H$_2$S, was diluted in water to calibrate the system at different concentrations of H$_2$S (0, 0.2, 1, 5, 25 and 125 μM) obtained from a stock solution of 125 mM (=500 mM of NaHS), knowing that H$_2$S corresponds to about 25% of the NaHS concentration[27]. The different concentrations of H$_2$S in the presence of SF7-AM were used to establish the standard curve which allowed to quantify the concentration of H$_2$S in predator secretions (anal gland scents for the skunk and urine for the others). The 96 well plates were read, after ~30–45 min at room temperature (RT), by a Wallac VICTOR3™ Microplate Reader and/or by The GloMax®-Multi Detection System (Promega). Spectrofluorimetric data were treated in Excel after exportation from the microplate reader. A minimum of two different plates were read ($n = 2$–6, depending on the secretions) and each plate was read three times. After subtraction of the intensity values of the controls (wells without SF7-AM for each condition), the H$_2$S concentration present in each secretion and/or in the control conditions (2-PT, TMT and SBT; $n = 2$ plates) was based on the equation of the standard curve. Values are expressed as mean + standard error of the mean (SEM).

We performed another control on H$_2$S solutions degassed using acidification[74]. Briefly, we added 500 μl of 5 M HCl to deplete a 5 ml stock solution of H$_2$S, which contained 2 mM of NaHS, corresponding to 500 μM of H$_2$S. This stock solution was left to degas for 48 h. The pH of this stock solution was then measured and adjusted with NaOH to a pH of 7.4. This stock solution was then used to prepare the different concentrations of H$_2$S (from 0.2 to 125 μM). The pHs of these experimental solutions were also measured to be 7.4. The same procedure was applied to the anal gland secretions from the skunk to obtain a solution of skunk H$_2$S-depleted.

### Animals
Adult C57BL/6 and OMP-GFP[32] mice (male and female) and pups (postnatal days P3–P8; for calcium imaging experiments) were used for these experiments. Mice were grouped-housed between 21 and 23 °C under a 12 h light/dark cycle with ad libitum access to food and water. They were killed by decapitation. The animal experiments in this study were approved by the EXPANIM committee of the Lemanique Animal Facility Network and the veterinary authority of Canton de Vaud (SCAV; VD2496.2, VD2496.3, VD2496.x3) in accordance with the Swiss legislation.

### Calcium imaging experiments on GG tissue slices
The precise protocol can be found in Brechbühl et al.[9]. Briefly, pup heads were placed, for the GG olfactory subsystem dissection, in ice cold artificial cerebrospinal fluid (ACSF: 118 mM NaCl, 25 mM NaHCO3, 10 mM D-glucose, 2 mM KCl, 2 mM MgCl$_2$, 1.2 mM NaH$_2$PO$_4$, 2 mM CaCl$_2$ (pH 7.4)) and gassed with oxycarbon (95% O$_2$, 5% CO$_2$). Tips of the noses were incorporated in 4% agar prepared in phosphate-buffered saline (PBS). GG tissue slices of 80 μm were then prepared on ice with a vibroslicer (VT1200S; Leica Biosystems, Welzlar, Germany). The slices were conserved in ACSF solution and selected under a fluorescent stereomicroscope (M165 FC,

Leica) to visualize GG neurons, thanks to their GFP expression. GG tissue slices were loaded with Fura-2AM (10 μM; Thermo Fisher Scientific) and pluronic acid (0.1%; InvitrogenTM F-127; Thermo Fisher Scientific) mixed in ACSF.

The slices were incubated for 45–60 min (37 °C, 5% $CO_2$) and then placed in the calcium imaging chamber (RC-26; Warner Instruments; USA). The GG tissue slices were continuously perfused with ACSF solution and observed with an inverted fluorescence microscope (ZEISS Axiovert 135) connected to a camera (Photometrics Scientific® CoolSNAP ES2 camera; Visitron Systems GmbH, Puchheim, Germany), which allowed for the computational visualization via the VisiView® (Visitron Systems GmbH, Germany) software. Observations were performed with the 25× objective at ex 480 nm/em 508 nm (GFP) and ex from 380 nm ($Ca^{2+}$ free) to 340 nm ($Ca^{2+}$-saturated) for the Fura-2 as it is a ratiometric fluorescent calcium indicator with an emission at 510 nm.

The GG neurons were stimulated with different $H_2S$ concentrations (1, 5, 12.5 and 25 μM) and compared to a TMT perfusion (100 μM). To test the viability of the neurons, ACSF-KCl (35 mM NaCl, 25 mM $NaHCO_3$, 10 mM D-glucose, 100 mM KCl, 3 mM $MgCl_2$, 1.2 mM $NaH_2PO_4$, 0.5 mM $CaCl_2$ gassed with oxycarbon 95% $O_2$ and 5% $CO_2$) was perfused at the beginning and end of each experiment.

Neurons that responded to the ACSF-KCl and generated responses from the cues that corresponded to at least 10% of the responses to ACSF-KCl (baseline activity) were selected to calculate the percentage of responding neurons and for the ratio amplitude of the responses.

To test the importance of extracellular calcium in the signaling pathway involved in danger recognition by GG neurons, we performed calcium imaging in the absence of extracellular calcium in the ACSF solution used while stimulating the neurons with the alerting cues.

For the analyses, the ratio amplitude responses were obtained dividing the magnitude of the amplitudes for each response to the cues by the magnitude of the amplitude to the KCl injection. Comparisons between experiments performed with normal ACSF and ACSF without calcium were done for each cue: 2-PT, TMT, SBT (2 mM) and $H_2S$ (25 μM).

To verify the effect of the gaseous portion of the solution, we prepared a depleted stock solution of $H_2S$ (500 μM) in ACSF adding HCl (5 M) as described previously in the *Spectrofluorimetry* section. We then perfused the $H_2S$ (25 μM), the $H_2S$-depleted (25 μM) and again the $H_2S$ (25 μM) solutions. The magnitude of the amplitude for each response of the perfusions were measured and divided by the magnitude of the amplitude of the KCl perfusion. A comparison of the two different conditions ($H_2S$ and $H_2S$-depleted) was performed. The same experiments were performed using the solution of skunk $H_2S$-depleted which was compared to the responses obtained with the skunk secretion on GG neurons.

### In vitro expression of CNGA3 channels

Human embryonic kidney (HEK) 293 cells were cultured in Dulbecco's Modified Eagle Medium (GibcoTM DMEM; Thermo Fisher Scientific, Waltham, MA, USA), supplemented with 0.2% gentamycin antibiotic (50 mg/mL; GibcoTM; Thermo Fisher Scientific, MA, USA) and 10% of foetal bovine serum (GibcoTM FBS; Thermo Fisher Scientific, MA, USA). Cells were cultured in T-25 flasks (Falcon®; VWR, Radnor, PA, USA) and incubated in a 5% $CO_2$ atmosphere at 37 °C.

HEK 293 cells were used for transfection, according to Moine et al.[22], with murine CNGA3 as gene of interest cloned in a mammalian expression vector (pcDNA3.1+/C-(K)-DYK; GenScript). Briefly, plasmid DNA was amplified and extracted from *Escherichia coli* bacteria. HEK 293 cells were cultured in 60 mm petri dishes (Falcon®; VWR, PA, USA) on a sterile microscope glass cover slip (22× 40 mm; Menzel ™; Thermo Fisher Scientific, MA, USA) for transfection. Cell media was changed and replaced with a new one 2–3 h before transfection. Cells were transfected via the calcium phosphate transfection method when the confluence reached 50–70%. Cells were co-transfected with the GFP reporter plasmid to verify the efficiency of the transfection. Cell transfection was also verified by immunocytochemistry experiments using THE™ DYKDDDDK Tag Antibody (GenScript;

Mouse; 1/500) against CNGA3 and Cy3 anti-mouse as secondary antibody (ThermoFisher; 1/200).

The cover slip with HEK 293 cells was transferred into the calcium imaging chamber. Cells were loaded with Fura-2 acetoxymethyl ester (AM) dye (10 μM) and pluronic acid (0.1%) mixed in Ringer solution (composed of 140 mM NaCl, 5 mM KCl, 10 mM HEPES, 2 mM $CaCl_2$ and 1 mM $MgCl_2$). Cells were incubated at 37 °C in the dark for 30–45 min. Cells were continuously perfused with Ringer solution and observed with an inverted fluorescence microscope (ZEISS Axiovert 135) connected to a camera (Photometrics Scientific® CoolSNAP ES2 camera; Visitron Systems GmbH, Puchheim, Germany), which allowed for the computational visualization via VisiView® (Visitron Systems GmbH, Germany) software. Observations were performed with the 25× objective at ex 480 nm/em 508 nm (GFP) and ex from 380 nm ($Ca^{2+}$ free) to 340 nm ($Ca^{2+}$-saturated) for the Fura-2 as it is a ratiometric fluorescent calcium indicator with an emission at 510 nm.

Adenosine 5′-triphosphate (ATP; 100 μM; Sigma-Aldrich® Merck, Schaffhausen, Switzerland) in Ringer solution was perfused via a syringe at the beginning and end of the experiments to verify the viability of the cells. Different concentrations of $H_2S$ were tested: 0.05, 0.25, 7.5, 10, 15, 20 and 25 μM. Pharmacological investigations were assessed with 8-Bromoguanosine 3′,5′-cyclic monophosphate (8-Bromo; 1000 μM; Sigma-Aldrich® Merck, Schaffhausen, Switzerland), an activator of CNGA[75] and with 3-isobutyl-1-methylxanthine (IBMX; 2, 10, 20, 40, 100 and 200 μM; Merck, Germany), a non-specific inhibitor of cAMP phosphodiesterases[7]. Other cues, 2-PT, SBT and TMT, were tested at 100 μM.

For the percentage of responding cells and the ratio amplitude of the responses, the calculation was made in the same way as for the GG neurons in the *Calcium imaging experiments on GG tissue slices* section, except that the magnitude of the amplitude was divided by the magnitude of the amplitude from ATP injection.

### S-sulfhydration assay

Lysate from CNGA3-transfected HEK cells was used for the colorimetric test[76]. After 24 h of transfection with the plasmid encoding CNGA3, cells grown in a petri dish were trypsinized and centrifuged in 5 mL of DMEM to harvest the cell pellet ($8 × 10^5$ cells; LUNA-II™ Automated cell counter, Bucher Biotec, Switzerland). As a control, the same pellet collection was performed with non-transfected cells. The cell pellet was then homogenized with 300 μL of RLT Plus lysis buffer (Qiagen GmbH, Germany; #10309). Lysate was then centrifuged at $12,000 × g$ for 10 min at 4 °C. 50 uL of the supernatant from non-transfected and CNGA3-transfected cells were added to a 96-well plate (Falcon®; VWR, Radnor, PA, USA). 5,5′-dithio-bis-(2-nitrobenzoic acid) (DTNB; 0.3 mM from stock solution 0.1 M prepared in PBS + EDTA 1 mM; Merck, Germany; #D218200) was used to test the sulfhydration reaction of $H_2S$ on CNGA3. Briefly, DTNB reacts with a free sulfhydryl group to yield a mixed disulfide and 2-nitro-5-thiobenzoic acid (TNB) the "colored" product with an absorbance at 412 nm[37,38]. Four different conditions were tested: non-transfected HEK cells +50 uL of DTNB + 50 uL Ringer solution; non-transfected HEK cells +50 uL DTNB and 50 uL of $H_2S$ (25 μM); CNGA3-transfected HEK cells 50 uL of DTNB + 50 uL Ringer solution and CNGA3-transfected HEK cells + 50 uL DTNB and 50 uL of $H_2S$ (25 μM). As negative control, iodoacetamide (IAA; 0.5 mM; Merck, Germany; #I6125), known to bind irreversibly free thiol groups[39], was used on non-transfected and CNGA3-transfected HEK cells. Each condition was done in duplicates. Plates were read at 412 nm (EPOCH2 Plate Reader; BioTek, Winooski, Vermont, USA) directly after preparation and over 100 min to obtain a kinetic curve of absorbance. Relative absorbance was then measured in arbitrary units over time for each condition (DTNB; DTNB + $H_2S$ and DTNB + IAA) represented as mean (of the duplicate) after subtraction of non-transfected HEK cells of each conditions.

### Stimulation of the Grueneberg ganglion (GG), olfactory bulb (OB) and brain regions (La, VMHDM and PAG)

To test the neuronal activity, mice were stimulated for 1 h with $H_2O$ (control) or $H_2S$ (125 μM), for the GG and for the OB, La, VMHDM and

PAG slices experiments. Briefly, blotting paper with 200 μL of H₂O or H₂S were added into the home cage of the mice. After stimulation, mice were sacrificed and the dissection of the GG, the OB and of the other brain regions were performed for further analysis.

## Immunohistochemistry on Grueneberg ganglion (GG) slices, olfactory bulb (OB) and brain regions (La, VMHDM and PAG)

Immunostainings were performed on the GG, the OB and the other brain regions (La, VMHDM and PAG) floating sections (VT1200S; Leica). Briefly, after a 4% PAF (paraformaldehyde prepared in a phosphate-buffered saline (PBS, pH 7.6)) fixation phase overnight (~12 h), mouse heads were transferred into a PBS solution. The GG, the OB and the brain regions were then precisely dissected and collected before being placed in 4% agar block.

Coronal tissue slices of 80 μm for the GG and 100–120 μm for the OB, La, VMHDM and PAG zones were then generated. Brain zones were selected according to Franklin and Paxinos (2001)[77].

The immunostaining procedure was then initiated by a blocking/ permeabilization phase with a specific serum (normal goat serum NGS; Interchim, France) and 1% Triton X-100 (TX100; Fluka analytical, Switzerland) for at least 2 h at RT. Primary antibodies were used in specific serum solution overnight at 4 °C: Rabbit anti-CNGA3 (1:800; Lifespan Bioscience), Rabbit anti-rpS6 (1/400; Cell signaling, MA, USA; #5364) and Rabbit anti-cFOS (1/500; ABCAM, Cambridge, UK; #ab190289). To identify the necklace glomeruli in the OB, a second primary antibody Rabbit anti-Pde1a was used (1:50; Thermo Fisher Scientific, MA, USA; PD1A-112AP) on the same tissue slices where the Rabbit anti-cFOS was previously tested. After washing phases, the following secondary antibody was used for 2 h at RT: Goat Cy3 anti-rabbit (1/200; Jackson ImmunoResearch, Cambridge, UK). Slice mounting was performed with Vectashield media containing DAPI (H-1200; Vector Lab, Servion, Switzerland).

Acquisitions were done by confocal microscopy (LEICA SP5 TANDEM, Leica Biosystems, Muttenz, Switzerland) with 40× objectives and analyzed with the computer assistance (v7.1.1, Imaris). Analysis for the intensity (rpS6 experiments) and of the number of cells (cFOS experiments) were performed with the Image J software. For the quantification of cFOS-positive cells, in the OB, more precisely the necklace glomeruli, the quantification was based on the number of cFOS+ cells considering a radius of 60 μm around each reconstituted glomerulus. For the other brain regions (La, VMHDM and PAG), a zone of 350 ×350 μm was used for the counting of the number of cFOS-positive cells. The average of a minimum of 3–10 slices was used for the establishment of the global number of cFOS-positive cells for the OB, La, VMHDM and PAG zones.

## Measurements of mice corticosterone level, blood pressure and heart rate

After stimulation of the mice with H₂S (125 μM), the plasma corticosterone levels were measured using the Corticosterone EIA Kit (Enzo Life Sciences). Briefly, mice were placed in contact with 200 μL of H₂O or H₂S on a blotting paper for 5 min in the home cage. Mice were then directly euthanized, and the blood samples were collected and centrifuged at room temperature for 5 min at 10,000 × g. Plasmas were kept at −80 °C until Elisa analysis. Corticosterone levels were measured in duplicate in accordance with the kit manufacturer.

For blood pressure measurements, the experiments were performed according to Brechbühl et al.[23], using the indirect and non-invasive computerized tail-cuff method. Briefly, mice were trained for the procedure during five continuous days with the equipment (BP-2000; Visitech) in a behavioral room (23 °C, normal light cycle) to limit experimental-related stress. Mice were placed in contact with H₂O or H₂S on a blotting paper, on the tail-cuff platform which is connected to a control unit and a computer. They were maintained in magnetic restrainers and their tails were placed in the cuff/pulse optical sensor. Diastolic and systolic pressures were perceived by monitoring the vessel dilation during the occlusion cuffs inflation (balloon inflation).

The diastolic pressure was defined as the cuff pressure necessary to observe the decrease of the waveform amplitude. The systolic blood pressure was defined as the cuff inflation pressure necessary to fall below 10% of its original stable amplitude. The mean pressure was calculated as the mean of the measured diastolic and systolic pressure. Measurements obtained in the presence of excessive animal movements (i.e., vigorous struggling or body twisting, indicating insufficient habituation or external stress) were discarded. For each automatized session, mean individual pressures (diastolic, systolic, and mean pressure) were obtained as the average of the 10 attempted measurements.

## Behavioral experiments

These experiments were performed according to Lopes et al.[15]. Briefly, an open field test was used to detect and observe fear-related behaviors of mice in the presence of the H₂S (25 or 125 μM). Mice were housed in the mouse facility with a 12:12 h light/dark inverted cycle in their home cage with food and water ad libitum.

One week before the beginning of the experiments, the animals were familiarized with the test arena to minimize the environmental stress. The test arena consisted of a closed Plexiglas box (45 × 25 × 19 cm) and 2 pieces of blotting paper (3 × 3 cm) in the middle of the width of the arena. The arena was divided in three different zones[46]: a safety zone, a central zone and a danger zone. A piece of blotting paper was positioned in the safety zone with a systematic addition of 1 ml of H₂O and another blotting paper was added in the danger zone with 1 ml of the tested cue (H₂O or H₂S at different concentrations). The presence of a blotting paper on both zones avoid any visual influence and focuses the test on the olfactory effect of the cues. In this arena, comparisons can be made between safety zone (H₂O) vs. danger zone (H₂O or H₂S). Blotting paper of the size of the box covered the floor to avoid direct contact of the mice with the Plexiglas.

Mice behaviors were recorded for at least 5 min during the nocturnal phase from the top of the arena covered with a Plexiglas plate by IR-sensitive HD camera under infrared vision. The recording session begun once the mice were introduced in the arena containing the blotting papers.

Video recordings were analyzed with ANY-maze software (Stoelting Europe, Dublin, Ireland), a video tracking system detecting the center of the animal as reference point, and the following parameters were quantified: the total distance travelled, the number of entries in the central zone, the distance travelled in the central zone and the number of entries in the danger zone and the total time of freezing.

Test sessions (H₂O vs. H₂O/H₂O vs. H₂S, 25 or 125 μM) were performed systematically at least 2–3 h after the control sessions (H₂O vs. H₂O) to avoid the order of cues exposure as a potential confounder. For each behavioral parameter, the data was processed in two steps to allow for comparisons across animals and experimental conditions[23]. First, for each mouse, the value obtained during the control session (H₂O vs. H₂O) was set to 100%, and the corresponding value from the test session (either H₂O vs. H₂O or H₂O vs. H₂S) was scaled proportionally. A difference index was then calculated for each individual by subtracting the normalized control value (100%) from the normalized test value, yielding a relative change in percentage. This normalization step accounted for inter-individual variability and differences in units between behavioral parameters. Secondly, we used the formula: $Index_i = (X_{test,i} - X_{control,i}) - \mu_{H_2O}$, where $X_{test,I}$ represents the normalized value obtained for the test, $X_{control,I}$ the normalized value obtained for the control session and $\mu_{H_2O}$ represents the mean of the reference H₂O, in order to center the H₂O distribution at 0% and to express the effect of H₂S relative to this baseline. Negative index values for distance-related parameters and number of entries indicate increased fear-related behaviors, such as reduced exploration. Positive index values for the freezing parameter reflect fear responses, as freezing is a well-established indicator of fear. The place of the safety zone was randomly changed between the mice.

## Axotomy procedure and behavioral assays

Generation of axotomized mice were performed according to Brechbühl et al.[9,33]. Briefly, C57BL/6 and OMP-GFP (male and female) mice, in which

axonal projection bundles of GG neurons were sectioned with 26 G needles under deep isoflurane anesthesia, were used for behavioral experiments 30 days post lesion. The axotomized mice were challenged in the same open field with stimulation of $H_2O$ (control) and $H_2S$ (25 or 125 µM) as previously described for normal mice (intact GG) in the *Behavioral experiments* section. Analysis of the behavioral assays were performed by an experimenter unaware of the phenotype of the mice.

To assess the success of the nerve axotomy, a floating immunohistochemical method was performed on GG tissue slices of normal mice and of axotomized mice (OMP-GFP and wild-type genotype; 80 µm) generated after the fixation with PAF 4%. For OMP-GFP mice, the presence of GG neurons was detected thanks to their GFP expression. For wild-type mice, an immunostaining was performed using as primary antibody the Goat anti-OMP (Wako; 1:1000). Briefly, slices were blocked for 3 h in a PBS solution containing 5% normal donkey serum (NDS; Jackson ImmunoResearch) and 0.5% triton X-100. The primary antibody was applied to the slices overnight in a PBS solution containing 2.5% NDS and 0.25% Triton X-100. Slices were then washed in 1% NDS and incubated with the secondary antibody (FITC-conjugated, Donkey anti-Goat; Jackson ImmunoResearch; 1:200) in 1% NDS for 2 h. Slice mounting was performed with Vectashield media containing DAPI (H-1200; Vector Lab, Servion, Switzerland). Acquisitions were made by confocal microscopy (SP5; Leica) with ×40 objectives and reconstructions were made with Imaris (Bitplane IMARIS 6.3). Only data from mice with full axotomy were considered.

To control the integrity of general odorant detection on axotomized mice, a cookie test with a buried Oreo cookie was performed after 24 h of food deprivation[55]. Briefly, the cookie was buried under 1 cm of bedding and the time to find de cookie was measured (in seconds) between normal and axotomized mice. Between each mouse, the location of the cookie varied.

## Statistics and reproducibility

GraphPad 9 (Prism®, San Diego, CA, USA) was used to generate aligned dots bar graphs and/or before-after graphs and to perform the statistical analyses. Sample size was determined on the basis of pilot experiments and according to previously reported publications in the field. Mice were not used randomly but in a controlled-attribution based on their genotypes, sex and age.

Statistical analyses were performed using a two-tailed paired Wilcoxon test for the ex vivo calcium imaging experiments comparing the ratio amplitude of the responses with and without calcium (ACSF *vs.* ACSF $Ca^{2+}$ free) and between $H_2S$ *vs.* $H_2S$-depleted.

A two-tailed Mann–Whitney test was used for the in vitro experiments using IBMX to compare the responses obtained with $H_2S$ on CNGA3-transfected cells (mean between before and after perfusions) and the responses obtained with $H_2S$ in the presence of the inhibitor. The same test was applied for the S-sulfhydration assay to compare the different conditions.

A two-tailed Mann–Whitney test was used for the experiments with rpS6 and a two-tailed Student's $t$ test for the experiments with cFOS for the two conditions ($H_2O$ *vs.* $H_2S$ stimulation).

A unidirectional test was used (one-tailed Mann–Whitney test) for the measurements of corticosterone.

A paired Student's $t$ test and a paired Wilcoxon test were used for the blood pressure and for the heart rate analysis, respectively, comparing the effects of control ($H_2O$ stimulation) with the ones of the cue ($H_2S$ stimulation).

A paired Student's $t$ test was also used to compare the behavioral control sessions ($H_2O$) and the test sessions ($H_2S$) for the in vivo experiments. Values are expressed as mean ± SEM with aligned dot plots. A two-tailed Student's $t$ test or a two-tailed Mann–Whitney was used for comparing the results of normal mice *vs.* axotomized mice for the different conditions. A two-tailed Mann–Whitney test was used for the cookie experiment.

No corrections were applied to compare multiple data. Significance levels are indicated as follows: *$p < 0.05$; **$p < 0.01$; ***$p < 0.001$; ns not significant.

## Reporting summary

Further information on research design is available in the Nature Portfolio Reporting Summary linked to this article.

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

## Acknowledgements

This work was supported by the Department of Biomedical Sciences at the University of Lausanne and by the Swiss National Science Foundation Grant 310030_185161 (to M.-C.B.). We thank Monique Nenniger Tosato, Zilan Yaren Susun, Yaëlle Wenger, Dean Wood as well as Chantal Verdumo for their expert technical help. We thank two facilities at the University of Lausanne: The Cellular Imaging Facility (CIF) and its coordinator Jean-Yves Chatton, The Electron Microscopy Facility (EMF) and Antonio Mucciolo for his help with the SEM image and Hélène Broillet for her volcanic photo. We would also like to thank Roland Bulliard at the Servion Zoo for allowing us to collect the predator secretions and capture their images. Elements of methods illustrations were created with BioRender.com (Fig. 1a, b, Fig. 3a, g, Fig. 4a, Fig. 5d, f).

## Author contributions

All authors reviewed and edited the manuscript. A.C.L., J.B. and M.-C.B. conceived the project and designed the experiments. A.C.L., J.B. and A.V developed the methods. A.C.L., J.B., A.V., N.G. and F.F. conducted the experiments and acquired the data. A.C.L, J.B., and A.V performed statistical and image analysis. A.C.L. and M.-C.B. wrote the paper.

## Competing interests

The authors declare no competing interests.
