## [Transparent Peer Review file · Communications Biology]

Hydrogen sulfide as a potent predator-derived kairomone mediating fear-related responses in mice

Corresponding Author: Professor Marie-Christine Broillet

This manuscript has been previously submitted at another journal. This document only contains information relating to versions considered at Communications Biology.

Version 1:

Reviewer comments:

Reviewer #1

(Remarks to the Author)

In their manuscript „Hydrogen sulfide as a potent predator-derived kairomone mediating fear-related responses in mice“ (# COMMSBIO-24-4865A), Lopes et al. use a variety of methods, such as axotomized mice, Calcium imaging experiments on Grueneberg ganglion (GG) tissue slices, immunohistochemistry on GG slices, olfactory bulb and other brain regions, measurement of physiological parameters, and behavioral studies, that may support their statements in and the title of their manuscript. Moreover, they measured effects of H₂S on HEK293 cells in which they expressed the CNGA3 channel.

General remarks

All in all, this is a nice and solid work with new findings of high interest in the field. The manuscript is easy to read, the experiments are state of the art and presented in a logical order. I do agree largely with the findings and statements of the authors inasmuch as the title of the manuscript is concerned. However, the manuscript, throughout, is flawed by unsubstantiated statements on a direct activation of CNGA3 channels by H₂S. For detailed criticism and alternative experimental approaches, see comments below.

Comments

1. Figure 3a: Nice staining, however, the expression of CNGA3 in GG neurons is not new, and in no context with the transfection of HEK293 cells. Subpanel (a) should go to the supplemental.

2. The authors rightfully claim that an interaction of H₂S with a variety of channels and transporters by a form of post-translational modification, the S-sulfhydration, has been reported previously (references 33 and 34) – however, CNGA3 or other CNG channels were not among those sulfhydrated proteins. Recently, Koike et al. (2021) suggested that in type B OSNs of the MOE H₂S inhibits PDEs, rather than directly activating a CNG channel (CNGA2) (Koike et al., Neuron 2021). Moreover, several groups suggested previously that the two transmembrane Guanyl Cyclase (GC) subtypes GC-D and GC-G expressed in a subset of sensory neurons in chemosensory tissues (MOE, GG) detect volatile compounds/alarm pheromones, such as guanylin and uroguanylin, 2-secbutyl-4,5-dihydrothiazole, as well as CO₂/bicarbonate and CS₂ (Hu et al, Science 2007; Leinders-Zufall et al, PNAS 2007; Duda & Sharma, BBRC 2008; Munger et al, Curr Biol 2010; Chao et al, Biochem J 2010, EMBO J 2018).

As it stands, in the present manuscript, Lopes et al. hypothesize but do not show a direct activation of CNGA3 by H₂S. Previous evidence rather suggested other proteins. For example, the results shown in Figure 3f may well be due to an inhibition of PDE (see Koike et al. Neuron 2021), since an inhibition of PDE will result in e.g. an increase in cyclic nucleotides by constitutive activity of endogenous adenylyl cyclase-activating GPCRs, which then will be detected by CNG channels. Moreover, the time kinetics of H₂S in Figure 3e are much slower than the kinetic of 8-bromo-cGMP (Fig. 3d), which also must pass the plasma membrane of HEK293 cells.

As of now, the Title of Figure 3 must be changed/toned down.

3. I suggest three experimental approaches that would improve the manuscript substantially.

1) Methods section, page 16, line 6ff and page 18, line 3ff, and Figure 3f: How long was the stimulation by H₂S? The authors should try the same experiment with increasing concentrations of IBMX instead of H₂S.

2) Lopes et al. speculate about protein S-sulfhydration of CNGA3. This is not shown by the authors. However, methods to detect protein sulfhydration have been published: D. Zhang et al., *Angew Chem Int Ed Engl.* 2014: „Detection of Protein S-Sulfhydration by a Tag-Switch Technique“; C. M. Park et al., *Methods Enzymol* 2015: „Use of the "tag-switch" method for the detection of protein S-sulfhydration“; for review, see Paul & Snyder, *TIBS* 2015 (Ref. 67). The manuscript would improve significantly, if the authors can experimentally demonstrate a S-sulfhydration of CNGA3.

3) Moreover, electrophysiological patch clamp experiments (excised patch!) with CNGA3-expressing HEK293 cells would definitely strengthen the manuscript.

In case these three experimental approaches are not possible for the authors, statements on a direct activation of CNGA3 by H₂S must be deleted or toned down throughout the manuscript.

4. In this context, as of now, the following statements must be deleted/rephrased, since they are not supported by the authors' experiments:

- Abstract, line 7: delete ‚H₂S activates the cyclic nucleotide-gated channels (CNGA3)‘
- page 7, line 8: deleted/replace the subheader
- delete the sentence on page 7, line 22
- delete the last part of the sentence on page 12, line 9 ff (...as we demonstrated,...‘)
- delete the sentence on page 12, line 21 („More precisely,...“)

Reviewer #2

(Remarks to the Author)

Detection of alerting substances from the environment, in particular alarm pheromones as well as kairomones released by predators, are of critical importance for the survival of many mammals. These chemical cues are (mainly) received via olfactory cells residing in distinct chemosensory compartments in the nose of mammals, including the Grueneberg ganglion (GG), a cluster of sensory neurons located in the nasal tip region that is crucial for eliciting fear-associated and avoidance behavior upon exposure to alarm pheromones or predator-derived kairomones.

In the present manuscript, Lopes and co-workers demonstrate that hydrogen sulfide (H₂S) is released in body secretions of carnivorous but not herbivorous mammals. More importantly, the authors show that H₂S activates GG neurons in mice in a dose- and Ca²⁺-dependent manner and as downstream neuronal processing centers also some areas in the murine brain known to be involved in mediating fear-/stress-related behaviors or responses. Intriguingly, responsiveness of GG neurons to H₂S seems to rely on the CNGA3 ion channel that has not been described previously as a chemosensory receptor in the olfactory system. Finally, H₂S-induced fearful behavior requires a functional GG, underscoring the importance of this tiny organ for hazard avoidance via detection of predator cues. Thus, in summary, the present manuscript provides a couple of novel and important findings with respect to the reception of the predator kairomone H₂S via the GG and the subsequent behavioral consequences, which are of interest for a broader readership. Since the experiments conducted in this study also appear to be sound, I would strongly support publication of this manuscript in *Communications Biology*, provided the authors are willing to address a number of major and minor issues listed below with respect to the presentation of the data and the description of given experiments.

Major aspects:

1) In several experiments, H₂S-depleted solutions are used for which H₂S depletion was achieved by adding the acid HCL. For this purpose, the authors added 500 µl of 5 M HCL to a H₂S-containing solution (page 15 line 5-7). Unfortunately, the amount of the H₂S solution supplemented with 500 µl of 5 M HCL is not given. Yet, can the authors show that the effects evoked by H₂S-depleted solutions are indeed based on H₂S depletion and not on lowering the pH value by adding HCL? If not, authors should tone down their conclusions for these experiments or even omit the relevant experimental approaches with H₂S-depleted solutions from the manuscript (including the corresponding conclusions presented in the Discussion section).

2) page 6 line 10: The sentence „... on GG tissue slices from OMP mice (Fig. 2b)“ is puzzling. What are OMP mice? Do the authors mean OMP-GFP mice?

3) page 7 line 8: In this headline (but also in some other text passages such as line 25 on page 7), it is claimed that H₂S activates the cyclic-nucleotide gated channel A3 (CNGA3) via S-sulfhydration. In fact, it is only shown that H₂S apparently activates this ion channel directly; however, no experiment is presented that convincingly indicates that this occurs via S-sulfhydration. Therefore, this claim should be deleted from the entire manuscript.

4) page 8 second paragraph and Fig. 4a-b and upper left panel in Fig. 4d: The major flaw of this manuscript is related to the experiments with the necklace glomeruli (NG). First, in the schematic representation shown in Fig. 4a, necklace glomeruli are indicated in only the dorsal region of the olfactory bulb. In fact, necklace glomeruli innervated by axons of GG neurons have been also found in other regions of the olfactory bulb, such as medial, lateral, and even ventral regions [as shown in detail by Bumbalo et al. 2017 [*Cell Mol Neurobiol* (2017) 37:729–742]]. Thus, the scheme in Fig. 4a and also the

corresponding text on page 8 are misleading. Second and more importantly, the images shown in Fig. 4b are labelled with NG, suggesting that the glomeruli shown are indeed necklace glomeruli. This (presumably wrong) impression is enhanced by the upper left panel in Fig. 4d, for which the relevant figure legend claims a “quantification of the density of cFOS+ cells observed after H₂O (white bars) and H₂S (grey bars) stimulation for NG”, and by a sentence in the discussion section (page 11 line 2) saying that “neuronal activity of the NG was indeed observed when the mice were stimulated with H₂S”. In this context, it has to be pointed out that necklace glomeruli are intermingled with other glomeruli in the olfactory bulb. As a result, in the absence of specific molecular markers for necklace glomeruli (as in this manuscript), it is (almost) impossible to discriminate between necklace glomeruli and other glomeruli. Consequently, it is very critical to claim that certain glomeruli are necklace glomeruli and consider these glomeruli for any kind of analysis if there is no clear evidence that these glomeruli indeed represent necklace glomeruli. Therefore, I strongly recommend to provide evidence that the glomeruli shown (Fig. 4b) and used for statistical analyses (Fig. 4d) are without any doubt necklace glomeruli. In any other case, these (then) misleading results should be omitted from the manuscript since they would lower the quality. Such dubious analyses with doubtful necklace glomeruli are not required because the authors convincingly demonstrate enhanced expression of the (neuronal) activity marker cFos upon exposure to H₂S in other relevant brain regions (Fig. 4c-d).

Finally, in the legend of Fig. 4b, the sentence “Representative cFOS activity (yellow), mature olfactory sensory neurons (green), nuclei stained with ...” is very problematic. What is shown in this figure is not cFos activity, but cFos immunoreactivity or expression. And there are no olfactory sensory neurons in the olfactory bulb; what is shown in green in Fig. 4b are instead axonal processes of olfactory neurons innervating glomeruli in the bulb.

5) page 10 line 20: It is claimed that “responses of GG neurons showed no desensitization”. Maybe I have overlooked the relevant experiments that substantiate this concept. However, from my point of view, the authors only show that increasing concentrations of H₂S lead to enhanced calcium signals in GG neurons when these cells were exposed to pulses of H₂S (Fig. 2b). In my opinion, to test for desensitization, the cells should be exposed to a given stimulus over a longer period of time or repeatedly exposed to the same stimulus (same H₂S concentration) with only rather short breaks between stimulations. Therefore, I suggest to remove the above-mentioned sentence.

6) page 16 line 21-23 and Fig. 2d-f: The authors say that “The ratio of the amplitude for each response of the perfusions were measured and divided by the amplitude of the KCl perfusion”. What is the ratio of the amplitude and how can you measure it? I assume that the authors mean that not the ratio, but the magnitude of the amplitude for each response of the perfusions was measured and divided by the magnitude of the amplitude of the KCl perfusion. Thus, they calculated the ratio. Correct? If my assumption is true, then it is also doubtful to give the relevant values in a. u. (arbitrary units?) as indicated in Fig. 2d-f. From a mathematical point of view, dividing a value in a given unit by another value in the same unit results in NO unit and not in an arbitrary one. Likewise, also for Figs. 3d and 3f, it might be problematic to use the a. u./arbitrary units since I assume that again the magnitude of the amplitude for each response of the perfusions were measured and divided by the magnitude of the amplitude of the perfusion with ATP (it is not specified in the manuscript that the response to ATP was used as a reference value). Correct?

Finally, this criticism my also apply to Fig. 3g as for this figure that is based on experiments with GG neurons, the authors do not specify in the figure legend what was used as a reference value. The response to KCl?

7) Throughout the entire manuscript, there is controversial information about the H₂S concentration used for the experiments. In the legend of Fig. 1 (page 28), it is said that “NaHS (H₂S donor) was used at different concentrations as scale reference (0, 0.2, 1, 5, 25 and 125 μM)”. In the Methods section (page 14), however, it is stated that “NaHS (Sodium hydrosulfide hydrate; Sigma-Aldrich; Merck, Germany), a donor of H₂S, was diluted in ddH₂O to calibrate the system at different concentrations of H₂S (0, 0.2, 1, 5, 25 and 125 μM)”. Therefore, it is unclear whether the above-mentioned concentrations (0, 0.2, 1, 5, 25 and 125 μM) refer to NaHS or to H₂S.

Moreover, it is also unclear how the authors know that H₂S corresponds to about 25% of the NaHS concentration (page 14 line 17). This is very confusing since it is hard to understand how the gaseous compound H₂S can correspond to the salt NaHS. In any case, for every experiment in this study, it should be explicitly specified whether the authors talk about the concentration of H₂S or NaHS.

8) Fig 1b and page 28 line 13: “A-C” should read “A + C” and “B-D” should read “B + D” (otherwise it is wrong and confusing)

9) Fig. 1e: What is indicated by the numbers on the x-axis? The H₂S or NaHS concentration in μM?

10) Fig. 2d: What do the light grey lines indicate/connect? Data points from the same cell in the presence and absence of Ca²⁺? Unfortunately, this is not mentioned in the figure legend (this also applies to Fig. 2e-f).

11) For the calcium imaging experiments with GG neurons, the number of analyzed cells is given in the legends of Figs. 2 and 3g. However, there is no information provided in these figure legends, indicating the number of animals (pups) from which these neurons were collected.

12) Fig. 2a-b: In the two micrographs depicted in Fig. 2a, OMP-specific staining is indicated. However, it is rather unlikely that this is true. If I am correctly informed, in the genome of the OMP-GFP mice used for this experiment, the OMP-encoding region was replaced by a sequence coding for the green fluorescent protein (GFP) (Potter et al. 2001; J Neurosci.; 21: 9713–9723.), raising the question how an OMP-specific staining is possible if there is no expression of OMP. I assume that “OMP” in this figure has to be replaced by “GFP”. This would be consistent with the relevant figure legend that claims that the green color denotes GFP expression.

The upper panel of Fig. 2b presumably also shows a tissue section through the GG of an OMP-GFP mouse (although this is not clear from the figure legend). If this notion is true, the respective figure legend (page 30 line 7-8) “Microscopy images of a tissue slice of a mouse GG observed with a DIC contrast (Hv) showing neurons in green (OMP) and after Fura-2 loading” should be better replaced by “Microscopy images of a tissue slice through the GG of an OMP-GFP mouse observed with a DIC contrast (Hv) showing GG neurons in green (GFP) and after loading with Fura-2”.

13) page 34 line 11 and Fig 4d: What means “quantification of the density of cFOS+ cells”? In Fig. 4d, there are values indicated, but no unit is provided for this density. In the Methods section (page 19 line 11), the authors only speak about the number of cells. Please specify.

14) The headline of the legend of Supplementary Fig. 1 (“Axo mice have no main odorant detection deficit ...”) requires revision. What do the authors mean with main odorant detection and what is a main odorant? Do the authors mean detection

of odorants via the main olfactory system? This heading is puzzling. I therefore suggest replacing “Axo mice have no main odorant detection deficit ...” with “Axo mice have no deficit in food detection ...”, as they are obviously not impaired in their search for food (a cookie).

Minor aspects:

15) For some experiments, sulfidefluor-7 acetoxymethyl ester (SF7-AM) is used as H₂S probe. What does that mean? Does SF7-AM bind to H₂S and does it change its fluorescence upon binding? It would be helpful for the reader to provide some relevant information about SF7-AM in the text (e.g. Methods section) and not only referring to a reference.

16) At several places throughout the manuscript (e.g. page 5 line 8) the authors use the term “animal secretions”. What do they mean exactly? Urine? Please specify.

17) The term “fisher” is used (page 5 line 9 and legend of Fig. 1) What does that mean? A (human) fisherman? In this context, for all the mammalian species employed in this study (mouse, elk, cattle, rabbit, raccoon, fox, fisher, mountain lion, and skunk), the correct scientific name(s) should be given as well (e.g. *Puma concolor* for the mountain lion) to avoid such misunderstandings.

18) page 6 line 5: What is GFP? Green fluorescent protein? Please explain this abbreviation.

19) page 6 lines 5-6: It is claimed that expression of GFP under the control of the promoter of the olfactory marker protein (OMP) allows the visualization of “all mature olfactory sensory GG neurons”. Has it ever been shown that the OMP promoter is active/activated in ALL mature GG neurons? Therefore, I recommend omitting the word “all”.

20) page 7 line 6: The sentence “These results demonstrated that H₂S acts as a kairomone via the activation of GG neurons ...” is not correct – at least not at this text passage. Up to this passage in the text, it has been only shown that H₂S is released by predators and activates GG neurons. Evidence that H₂S actually acts as a kairomone by triggering certain physiological and behavioral responses is provided later in the text. Therefore, the relevant sentence should read “These results demonstrated that H₂S activates GG neurons”.

21) page 7 lines 14-15: At this, but also at other text passages, the wording regarding proteins that are expressed in human cells using plasmid vectors should be improved. Consequently, the sentence “... in which we transfected a reporter plasmid, GFP and the murine CNGA3 plasmid” should better read “... in which we transfected a reporter plasmid encoding GFP and a second plasmid coding for the murine CNGA3 channel”. At least, I assume that the authors did not transfect cells with a reporter plasmid, GFP, and the murine CNGA3 plasmid (what is a murine plasmid?) but with two plasmids of which one encodes GFP and the other the murine CNGA3 channel. Please consider this aspect also for other relevant passages of the manuscript (such as the legend of Fig. 3).

22) The description of cells lacking or being endowed with CNGA3 or cFos should be improved (e.g. page 7 line 19, page 8 line 11, legend of Fig. 3). For instance, in the legend of Fig. 3, a “CNGA3 – cell” is mentioned. It is not immediately clear whether this cell expresses or lacks CNGA3. Therefore, it would be better to designate “– cells” as CNGA3-negative (or cFos-negative) and “+ cells” as CNGA3-positive/expressing (or cFOS-positive/expressing) cells (throughout the entire manuscript).

23) page 8 line 14: The sentence “... implicated in the treatment of danger information ...” should read “... implicated in the processing of danger information ...”

24) page 10 line 4: In the sentence “... behaviors in the preys such as the mice” should read “... behaviors in mice” since we do not know whether H₂S also elicits similar responses in other prey animals, do we?

25) page 11 line 11-12: The sentence “Thus, H₂S present in the predator secretions could induce in the prey this artificial hibernation as a survival strategy” is very puzzling. Has hibernation ever been observed in *Mus musculus*? Presumably, it is better to omit this sentence from the manuscript.

26) page 14: The authors describe the spectrofluorimetry approach used to quantify the H₂S concentration in several solutions and predator secretions. In the description of this experiment, it was not clear to me what they used the fluorescence stereomicroscope (Leica M165 FC) for because visualization of the H₂S concentration is not shown (at least not in micrographs). Moreover, in this paragraph, the authors should explain the abbreviations ddH₂O (double distilled water?) and RT (room temperature?).

27) page 15 line 9: For the OMP-GFP mice, the reader is referred to reference 30. As far as I know, this is not the original publication in which the generation of this transgenic mouse line has been initially described and characterized. Thus, it would be desirable that the reader is referred to the original publication.

28) With respect to the latter aspect, it is a flaw of the present manuscript that in particular regarding the GG, its molecular phenotype, and its ability to detect alerting compounds, the reader is only referred to a few publications that are mostly from the authors' own group. For instance, concerning the localization of CNGA3 in cilia of GG neurons, the reader is referred to reference 32 (a publication of the authors' own group). However, this observation has been initially published several years before by another group (Liu CY, Fraser SE, Koos DS. Grueneberg ganglion olfactory subsystem employs a cGMP signaling pathway. *J Comp Neurol* 516: 36-48, 2009). Thus, it seems that the authors are only aware of their own publications. In my opinion, such self-citation and neglect of the work of others does not meet the high scientific standards given by this journal. Authors should therefore consider to cite the work of others at appropriate passages in the text.

29) page 16 line 4 and page 17 line 24: The sentence “observations were performed at 480 nm (GFP) and 380 nm (Fura-2) ...” is not very precise. I assume that the given wavelengths (480 nm for GFP and 380 nm for Fura-2) denote the excitation wavelengths, don't they? What about the emission wavelength(s)? Please specify.

30) page 18 line 1: “Adenosine 5'-triphosphate in Ringer solution (100 μM; Sigma-Aldrich @ ATP; Merck, Schaffhausen, Switzerland) was perfused ...” should read “Adenosine 5'-triphosphate (ATP; 100 μM; Sigma-Aldrich Merck, Schaffhausen, Switzerland) in Ringer solution was perfused ...”

31) page 18 line 19: Which buffer was used to dissolve the 4% PAF? How long did the fixation step take?

32) page 19 line 14: How long were the mice exposed to H₂S?

33) page 20 line 6-7: What means “excessive animal movements”?

34) page 20 line 18: When did the recording phase of 5 min start? Immediately after placing the blotting paper treated with H₂O or H₂S?

35) page 21 line 13-14: "... GG neurons was detected thanks to their OMP expression" should read "... GG neurons was detected thanks to their GFP expression"

36) page 28 line 7: It would be better to replace "cut" by "section".

37) Fig. 3b: "24h 48" should read "24-48 h"

38) page 32 line 16: I recommend to replace "percentage of responding cells (left)" by "percentage of responding HEK cells (left)" to avoid any confusion with GG cells for which the results are shown in the next figure (Fig. 3g).

39) Fig. 5c: What means BPM? Beats per minute?

40) Figs. 5e and 5g and Supplemental Fig. 1b: "Freezing" should be placed by "Time of freezing".

Moreover, in these figures, the presentation of the data regarding the standardized index is somewhat unusual and its calculation not entirely clear (since it has not been described in the manuscript). Let's assume that intact control mice exposed to H₂O have covered a total distance of 1 meter (on average) within the period of observation (Fig. 5e). For comparison with other intact mice exposed to H₂S, wouldn't it make sense to set this (average) total distance of 1 meter at 100% and not 0% (as the authors apparently did)?

For Fig. 5g, it is unclear which mouse was compared with which mouse. I assume that Ctrl mice exposed to H₂O were compared to Ctrl mice exposed to H₂S. And Axo mice exposed to H₂O were compared to Axo mice exposed to H₂S. Furthermore, I assume that the values for the Ctrl mice in Fig. 5g were taken from the experiment(s) depicted in Fig. 5e. If this assumption is correct, then the authors should describe it like that in the corresponding figure legend. Their description "and by the comparison of the different parameters (g) previously measured in (e)" is difficult to understand for the reader because it is unclear what comparison of the different parameters means.

In Supplementary Fig. 1b, it is not clear which value was set as the reference (i.e. 0%). I suspect that Ctrl mice exposed to H₂O were used as a reference for comparison with the mice in the other experimental conditions. Therefore, the authors should clearly describe this in the figure legend.

Version 2:

Reviewer comments:

Reviewer #1

(Remarks to the Author)

The effort of the authors to perform necessary, additional experiments has improved the manuscript substantially. I have only few comments on the revised manuscript:

1. Fig. S2b, d Use the same font size for x-axis labeling (preferably the larger font!)
2. Fig. S2b,d About 20% responding cells at 200 mM IBMX makes sense, and reasonably depicts a transfection efficiency. In this context it is not clear what '100% responding cells' in Fig. S2d, left panel mean. Does it mean that all cells that responded to H₂S in IBMX-Ringer also responded to H₂S in Ringer without H₂S? Please clarify.
3. Fig. S2a, c The vertical scale bars in panels a, c should be labelled with ratio numbers, as done in b, c (right panels). I suppose there are no units ([au]) since this is a FURA-2 ratio of 515 nm emission signals induced by stimulation with either 340 or 380 nm...

Reviewer #2

(Remarks to the Author)

The manuscript by Lopes and co-workers was significantly improved by the changes made by the authors. In particular, the authors have addressed many of the critical points raised by the reviewers. Therefore, I can endorse the publication of this manuscript. However, there are still some major and minor aspects that require additional attention and should be clarified before publication.

Major aspects:

1. page 9 line 14-18: The sentences "We observed a significant increase in the number of cFOS-positive nuclei after stimulation with H₂S (+69 % of activation) compared to the baseline activation observed with H₂O stimulation in one represented glomeruli (Fig. 4b and d). cFOS+ NG were also systematically tested for the phosphodiesterase 1a (Pde1a) expression, a specific marker of the NG45 (Supplementary Fig. 3)." are difficult to understand. Was only one glomerulus actually analyzed? Probably not. Thus, I assume, it should read "Compared to the baseline activation observed with H₂O stimulation, we observed a significant increase in the number of cFOS-positive cells (+69 %) surrounding glomeruli after stimulation of animals with H₂S (Fig. 4b and d). For this analysis, only glomeruli were considered that were immunoreactive for phosphodiesterase 1a (Pde1a), a specific marker of the NG45 (Supplementary Fig. 3)." Is that what the authors wanted to convey? Furthermore, the figure of 69 % cannot be correct because the bar for H₂S is almost three times as large as that for H₂O. Consequently, there should be an increase of about 200%. Importantly, another aspect remains puzzling: the authors claim in their rebuttal letter that for the revised version of the manuscript, they have only analyzed glomeruli positive for the necklace glomeruli (NG) marker Pde1a. This does not appear to have been the case for the initial version of the manuscript in which cFos-expressing cells were counted regardless whether the relevant glomerulus was Pde1a-positive or Pde1a-negative (at least a co-staining with an antibody for Pde1a was not mentioned in the original version). In this context, it is very surprising that the data set (i.e. the numbers in the bar chart in Fig. 4d) presented for this experiment appears to be the same as in the original version of the manuscript.

2. page 16 line 16-17 and page 18 line 13-14: It seems that the excitation wavelength used for calcium imaging experiments varies between calcium-free (380 nm) and calcium-saturated (340 nm) conditions. Why is this so? Presumably, this could influence the experimental results. Can these results still be compared (for instance in Fig. 2c) if the excitation wavelength was not identical?

3. Page 34 line 8: Were the 11 neurons analyzed in Fig. 2f actually from only one pup? Is that representative?

4. Fig. 5e + 5g + Suppl. Fig 4b (including the corresponding figure legends and the description of these results in the methods section): The calculation of the "index standardized [%]" is still absolutely elusive. The authors say in their rebuttal letter that "Control mice were first exposed to H₂O vs. H₂O in what we called the control session, during which they covered a total distance of approximately 1 meter. The test session was performed 2–3 hours later. In this session, control mice were again exposed to H₂O vs. H₂O, while another group of control mice was exposed to H₂O vs. H₂S. For the mice tested with H₂O vs. H₂O, in both sessions, their total distance covered remained similar between sessions (i.e., around 1 meter in each session). As a result, the difference between the test session and the control session was close to 0, and we standardized this difference to 0% (represented as white bars in Fig. 5e)." Consequently, if in a H₂O vs. H₂O approach, the difference between the test session and the control session was close to 0 meter, and the authors standardized this difference to 0%, this would mean that a difference of 0 meter was defined as 0%. Right? For the mice exposed to H₂O vs. H₂S, in the test session, the authors claim that "their total distance covered was reduced compared to their own control session (H₂O vs. H₂O). This decrease resulted in a negative index, as shown by the grey bars in Fig. 5e." Let us assume that these animals also covered 1 m under control conditions (H₂O versus H₂O) but only 0.6 meter under test conditions (H₂O versus H₂S). In this scenario, the difference between the test and the control session would be -0.4 meter. If 0 meter correspond to 0% (as mentioned by the authors), what would be the corresponding percentage for -0.4 meter? From my point of view, this type of calculation does not make sense. Therefore, the authors should provide a clear, comprehensive and mathematically correct formula for calculating this so-called "index standardized [%]". Such a formula should not only be given for the total distance, but also for entries central zone, distance central zone, entries danger zone and time of freezing.

Minor aspects:

5. page 7 line 1: "2-PT, -100 %; TMT, 45 % and H₂S, -24 %" should read "2-PT, -100 %; TMT, -45 % and H₂S, -24 %" (the minus sign in relation to 45% should not be missed)

6. page 11 line 23-25: It is not at all surprising that increased concentrations of the stimulus (H₂S) lead to enhanced behavioral responses. Initially, this has nothing to do with a lack of desensitization. Therefore, in the sentence "Interestingly, in behavioural experiments, increasing the concentration of H₂S induced accentuated fear-related responses suggesting a lack of desensitization." should read "In behavioural experiments, increasing the concentration of H₂S also induced accentuated fear-related responses."

7. with respect to point 1 of my initial referee report (page 15 line 13-15 of the present manuscript): The authors describe that to deplete the stock solution of H₂S, they added 500 µl of 5 M HCl to a stock solution, which contained 2 mM of NaHS, corresponding then to 500 µM of H₂S. Unfortunately, the volume of the NaHS stock solution to which 500 µl of 5 M HCl was added is not mentioned. Was it 10 ml or 100 ml or how much of NaHS solution?

8. page 18 line 15: As already mentioned in my last referee report "Adenosine 5'-triphosphate in Ringer solution (100 µM; Sigma-Aldrich @ ATP; Merck, Schafftrausen, Switzerland) was perfused ..." should read "Adenosine 5'-triphosphate (ATP; 100 µM; Sigma-Aldrich Merck, Schaffhausen, Switzerland) in Ringer solution was perfused ...".

9. page 19 line 3: The "the plasmid CNGA3" should read "the plasmid encoding CNGA3".

10. legend of Fig. 1b (page 31 line 15-16): The authors claim that micrographs acquired with a fluorescence microscope were added to a schematic representation of the plastic plate. What do the authors mean with "micrographs"? These dots in different shades of green on the right-hand panel of Fig. 1b? By definition, a micrograph is an image, captured photographically (or digitally), taken through a microscope to show a magnified image of an object. I'm sorry, but I can't recognize a micrograph in Fig. 1b. But maybe, I missed it.

11. coming back to point 9 of my initial referee report: The description of the x-axis in Fig. 1e (page 31) is still confusing. In their rebuttal letter, the authors claim that all concentrations given are the concentrations of H₂S. This would mean that both the values on the x-axis and on the y-axis would denote the H₂S concentration. Does that make sense? In my opinion, no. If I understood the experiment correctly, the authors determined the H₂S concentrations of NaHS solutions. Wouldn't it therefore make more sense to show the NaHS concentration of these solutions on the x-axis and the resulting H₂S concentration of the respective solution on the y-axis?

12. legend of Suppl. Fig. 1: "CNGA3 channels are expressed at the cilia ..." should read "CNGA3 channels are located in the cilia ...".

13. both bar charts in Suppl. Fig. 2d and the corresponding figure legend (line 27): Wouldn't it make more sense to replace "H₂S-IBMX" by "H₂S + IBMX", as this solution contains H₂S plus IBMX (obviously, it is not H₂S minus IBMX)?

Version 3:

Reviewer comments:

Reviewer #2

(Remarks to the Author)

The authors have (for the most part) adequately addressed the points of criticism raised by this reviewer. However, there is still one aspect that urgently needs clarification. Otherwise, I can recommend the manuscript for publication in this journal.

Major aspect:

1. With respect to point 4 of my last referee report, for Fig. 5e + 5g + Suppl. Fig 4b (including the corresponding figure legends and the description of these results in the methods section), the calculation of the “index standardized [%]” is still absolutely elusive. The authors present the formula: $Index_i = (X_{test,i} - X_{control,i}) - \mu H_2O$. These abbreviations are totally unclear and should be explained. What does μH_2O stand for, for example? And how does such a formula lead to a value given in %?

In addition, the description of the experimental procedure of the relevant behavioral experiments in the methods section is extremely sparse and the literature references provided are not very helpful. In this context, the authors wrote in their recent rebuttal letter that “Control mice were first exposed to H2O vs. H2O in what we called the control session.(...). The test session was performed 2–3 hours later. In this session, mice were again exposed to H2O vs. H2O, while another group of mice was exposed to H2O vs. H2S. This important information is only given very briefly in the methods section. It is therefore very difficult for the reader to understand the experimental set-up. I suggest describing the (entire) procedure of the behavioral experiments in much more detail; in a way that is also clearly comprehensible for the reader.

Hydrogen sulfide as a potent predator-derived kairomone mediating fear-related responses in mice

Responses to the reviewers:

Reviewer #1

General remarks

We would like to thank the reviewer for his/her very constructive remarks, for the very appreciated comments and for the detailed criticism. Thank you also for the alternative experimental approaches proposed.

Comments

1. *Figure 3a: Nice staining, however, the expression of CNGA3 in GG neurons is not new, and in no context with the transfection of HEK293 cells. Subpanel (a) should go to the supplemental.*

We thank the reviewer for this very pertinent comment. We have thus decided, as proposed, to move the subpanel (a) from Figure 3 to the supplemental. The immunostainings on GG neurons showing the CNGA3 in cilia of GG neurons are now the **Supplementary Fig. 1**.

2. *The authors rightfully claim that an interaction of H₂S with a variety of channels and transporters by a form of post-translational modification, the S-sulfhydration, has been reported previously (references 33 and 34) - however, CNGA3 or other CNG channels were not among those sulfhydrated proteins. Recently, Koike et al. (2021) suggested that in type B OSNs of the MOE H₂S inhibits PDEs, rather than directly activating a CNG channel (CNGA2) (Koike et al., Neuron 2021). Moreover, several groups suggested previously that the two transmembrane Guanylyl Cyclase (GC) subtypes GC-D and GC-G expressed in a subset of sensory neurons in chemosensory tissues (MOE, GG) detect volatile compounds/alarm pheromones, such as guanylin and uroguanylin, 2-secbutyl-4,5-dihydrothiazole, as well as CO₂/bicarbonate and CS₂ (Hu et al, Science 2007; Leinders-Zufall et al, PNAS 2007; Duda & Sharma, BBRC 2008; Munger et al, Curr Biol 2010; Chao et al, Biochem J 2010, EMBO J 2018).*

As it stands, in the present manuscript, Lopes et al. hypothesize but do not show a direct activation of CNGA3 by H₂S. Previous evidence rather suggested other proteins. For example, the results shown in Figure 3f may well be due to an inhibition of PDE (see Koike et al. Neuron 2021), since an inhibition of PDE will result in e.g. an increase in cyclic nucleotides by constitutive activity of endogenous adenylyl cyclase-activating GPCRs, which then will be detected by CNG channels. Moreover, the time kinetics of H₂S in Figure 3e are much slower than the kinetic of 8-bromo-cGMP (Fig. 3d), which also must pass the plasma membrane of HEK293 cells. As of now, the Title of Figure 3 must be changed/toned down.

We would like to thank the reviewer for this relevant question. Indeed, Koike et al. (2021) suggested that, in type B OSNs of the MOE, H₂S inhibits PDEs, rather than directly activating a CNG channel (CNGA2). We decided now to address this issue experimentally, in our *in vitro* model. We used, as suggested by the reviewer, IBMX, as

a PDE inhibitor, and we observed that, despite the presence of this inhibitor, H₂S still induces intracellular calcium increases in CNGA3-transfected HEK cells. We found that there is no significant difference between the number of responding cells and the amplitude of the responses with and without IBMX. Therefore, the observed H₂S effect is not via the inhibition of PDE and further modification of cyclic-nucleotide concentration. We now show these additional results in a new **Supplementary Fig. 2c-d**.

Concerning the response to 8-bromo-cGMP, we apologized for first selecting a non-representative experimental response. We now have changed to a representative intracellular calcium response in a new **Fig. 3c**.

3. I suggest three experimental approaches that would improve the manuscript substantially.

1) Methods section, page 16, line 6ff and page 18, line 3ff, and Figure 3f: How long was the stimulation by H₂S? The authors should try the same experiment with increasing concentrations of IBMX instead of H₂S.

We thank the reviewer for suggesting this additional experiment. Accordingly, we have performed calcium imaging experiments on CNGA3-positive cells perfusing a gradient of IBMX. We observed no intracellular calcium increase in these conditions. These new results are now shown in the new **Supplementary Fig. 2a-b**. We also performed H₂S perfusions in the presence of IBMX and observed no significant differences. We thus confirm that H₂S-related responses are not dependant of PDE inhibition.

2) Lopes et al. speculate about protein S-sulfhydration of CNGA3. This is not shown by the authors. However, methods to detect protein sulfhydration have been published: D. Zhang et al., Angew Chem Int Ed Engl. 2014: „Detection of Protein S-Sulfhydration by a Tag-Switch Technique“; C. M. Park et al., Methods Enzymol 2015: „Use of the "tag-switch" method for the detection of protein S-sulfhydration“; for review, see Paul & Snyder, TIBS 2015 (Ref. 67). The manuscript would improve significantly, if the authors can experimentally demonstrate a S-sulfhydration of CNGA3.

We thank the reviewer for suggesting this experimental approach to demonstrate protein S-sulfhydration of CNGA3. While the “tag-switch” method would have been a very interesting experiment to do, we finally had to use, because of the lack of accessibility to the tag-switch chemical, another approach, a “colorimetric” assay using the properties of the DTNB molecule known to bind free SH- groups to demonstrate the S- sulfhydration of CNGA3 (Ellman (1959); Winther et al. (2014); Broillet et al. (1996)).

In this assay, the reaction of DTNB (5,5'-dithio-bis-(2-nitrobenzoic acid)) with free SH- groups lead to the production of mixed disulfides and 2-nitro-5-thiobenzoic acid (TNB), a yellow-coloured product with an absorbance at 412 nm (new **Fig. 3g**; left panel). When DTNB was bound to CNGA3 free cysteine residues, we observed an increase in the relative absorbance over time, meaning that CNGA3 indeed possess free SH- groups (new **Fig. 3g**; right panel). Then, adding H₂S as a competitor of DTNB decreased the production of TNB as we observed a significant decrease in the relative absorbance (Mann-Whitney two-tailed test; p=0.0022). This also suggests that H₂S

binds to the free SH- groups of CNGA3 in a reversible manner. As a negative control for our assay, we tested the competition between DTNB and iodoacetamide (IAA), an irreversible ligand to SH- groups and observed no relative absorbance (**Fig. 3g**). Relative absorbance of the negative control was significantly different compared to DTNB condition and DTNB + H₂S condition (Mann-Whitney two-tailed test; p=0.0022). These new results are shown in the **new Fig. 3g**. We have now experimentally demonstrated the S-sulfhydrylation of CNGA3.

Fig. 3g - Fluorimetric biochemical assessment of CNGA3 S-sulfhydrylation by H₂S. Schematization of DTNB effect on the free thiol (SH-) groups performed on a lysate of HEK cells expressing CNGA3 (on the left). Measurement of relative absorbance (arbitrary units; a.u) of TNB over time in minutes at 412 nm (on the right). Three different conditions were tested : DTNB alone (0.3 mM; black line); DTNB + IAA (0.5 mM; magenta line) and DTNB + H₂S (25 μM; orange line).

3) Moreover, electrophysiological patch clamp experiments (excised patch!) with CNGA3-expressing HEK293 cells would definitely strengthen the manuscript.

We thank the reviewer for suggesting three different experimental approaches to demonstrate the S-sulfhydrylation of CNGA3 channels and we now have successfully used two of them. These added pharmacological and biochemical assays complement nicely the methodology previously used in the paper. Additional electrophysiological single-channel patch-clamp experiments on CNGA3-expressing HEK293 could indeed further confirm the direct activation, as a third method and we now mentioned this in the discussion of the revised manuscript.

In case these three experimental approaches are not possible for the authors, statements on a direct activation of CNGA3 by H₂S must be deleted or toned down throughout the manuscript.

4. In this context, as of now, the following statements must be deleted/rephrased, since they are not supported by the authors' experiments:

- Abstract, line 7: delete, H₂S activates the cyclic nucleotide-gated channels (CNGA3)'
- page 7, line 8: deleted/replace the subheader
- delete the sentence on page 7, line 22
- delete the last part of the sentence on page 12, line 9 ff (...as we demonstrated,...')
- delete the sentence on page 12, line 21 (,More precisely,...)

We thank the reviewer for these suggestions of modifications. However, we did not need to delete/rephrase these statements as, thanks to the reviewer's methodology suggestions, we have now provided new experimental results which support our hypothesis that H₂S activates directly CNGA3 (in an *in vitro* model) by S-sulfhydration (**new Fig. 3g**).

Hydrogen sulfide as a potent predator-derived kairomone mediating fear-related responses in mice # COMMSBIO-24-4865A

Responses to the reviewers:

Reviewer #2

General remarks

We would like to thank the reviewer for his/her very constructive and useful comments. We took care of all his/her requested modifications, and we think that the manuscript is now greatly improved.

Major aspects

1) *In several experiments, H₂S-depleted solutions are used for which H₂S depletion was achieved by adding the acid HCL. For this purpose, the authors added 500 µl of 5 M HCL to a H₂S-containing solution (page 15 line 5-7). Unfortunately, the amount of the H₂S solution supplemented with 500 µl of 5 M HCl is not given. Yet, can the authors show that the effects evoked by H₂S-depleted solutions are indeed based on H₂S depletion and not on lowering the pH value by adding HCl? If not, authors should tone down their conclusions for these experiments or even omit the relevant experimental approaches with H₂S-depleted solutions from the manuscript (including the corresponding conclusions presented in the Discussion section).*

We apologize for the lack of details provided in our original experimental protocol and we have now modified it accordingly. To deplete the stock solution of H₂S, we added 500 µl of 5 M HCl to a stock solution, which contained 2 mM of NaHS, corresponding then to 500 µM of H₂S. This stock solution was left to degas for 48 hours. The pH of this stock solution was then measured and adjusted with NaOH to a physiological pH of 7.4. This stock solution was then used to prepare the different concentrations shown in the different figures (from 0.2 to 125 µM H₂S). The pHs of these experimental solutions were also measured to be 7.4. We now added this detailed procedure on page 15 line 13-19.

2) *page 6 line 10: The sentence "... on GG tissue slices from OMP mice (Fig. 2b)" is puzzling. What are OMP mice? Do the authors mean OMP- GFP mice?*

We apologize for the confusion, and we have now changed the sentence on page 6, line 13 of the new manuscript. Indeed, OMP mice are OMP-GFP mice.

3) *page 7 line 8: In this headline (but also in some other text passages such as line 25 on page 7), it is claimed that H₂S activates the cyclic-nucleotide gated channel A3 (CNGA3) via S-sulphydration. In fact it is only shown that H₂S apparently activates this ion channel directly; however no experiment is presented that convincingly indicates that this occurs via S-sulphydration. Therefore, this claim should be deleted from the entire manuscript.*

This is indeed an important issue and the reviewer #1 proposed new experimental approaches to demonstrate the S-sulphydration of CNGA3. We decided to perform these experiments which finally confirmed our hypothesis, and we have now included these new results in a new **Fig. 3g**. In summary, we used a "colorimetric" assay using

the properties of the DTNB molecule known to bind free SH- groups (Ellman (1959); Winther et al. (2014); Broillet et al. (1996)).

The reaction of DTNB (5,5'-dithio-bis-(2-nitrobenzoic acid)) with free SH- groups lead to the production of mixed disulfides and 2-nitro-5-thiobenzoic acid (TNB), a yellow-coloured product with an absorbance at 412 nm (new **Fig. 3g**; left panel). When DTNB was bound to CNGA3 free cysteine residues, we observed an increase in the relative absorbance over time, meaning that CNGA3 indeed possess free SH- groups (new **Fig. 3g**; right panel). Adding H₂S as a competitor of DTNB decreased the production of TNB as we observed a significant decrease in the relative absorbance (Mann-Whitney two-tailed test; p=0.0022). This also suggests that H₂S binds to the free SH- groups of CNGA3 in a reversible manner. As a negative control for this assay, we tested the competition between DTNB and iodoacetamide (IAA), an irreversible ligand to SH- groups and observed no relative absorbance (**Fig. 3g**). Relative absorbance of the negative control was significantly different compared to DTNB condition and DTNB + H₂S condition (Mann-Whitney two-tailed test; p=0.0022). These new results are shown in the new **Fig. 3g** and experimentally demonstrate the S-sulfhydration of CNGA3.

Fig. 3g - Fluorimetric biochemical assessment of CNGA3 S-sulfhydration by H₂S. Schematization of DTNB effect on the free thiol (SH-) groups performed on a lysate of HEK cells expressing CNGA3 (on the left). Measurement of relative absorbance (arbitrary units; a.u) of TNB over time in minutes at 412 nm (on the right). Three different conditions were tested : DTNB alone (0.3 mM; black line); DTNB + IAA (0.5 mM; magenta line) and DTNB + H₂S (25 μM; orange line).

4) page 8 second paragraph and Fig. 4a-b and upper left panel in Fig. 4d: The major flaw of this manuscript is related to the experiment with the necklace glomeruli (NG). First, in the schematic representation shown in Fig. 4a, necklace glomeruli are indicated in only the dorsal region of the olfactory bulb. In fact, necklace glomeruli innervated by axons of GG neurons have been also found in other regions of the olfactory bulb, such as medial, lateral, and even ventral regions [as shown in detail by Bumbalo et al. 2017 [Cell Mol Neurobiol (2017) 37:729-742]]. Thus, the scheme in Fig. 4a and also the corresponding text on page 8 are misleading. Second and more importantly, the images shown in Fig. 4b are labelled with NG, suggesting that the glomeruli shown are indeed necklace glomeruli. This (presumably wrong) impression is enhanced by the upper left panel in Fig. 4d, for which the relevant figure legend claims a "quantification of the density of cFOS+ cells observed after H₂O (white bars) and H₂S (grey bars) stimulation for NG", and by a sentence in the discussion section (page 11 line 2) saying that "neuronal activity of the NG was indeed observed when the mice were stimulated with H₂S". In this context, it has to be pointed out that

necklace glomeruli are intermingled with other glomeruli in the olfactory bulb. As a result, in the absence of specific molecular markers for necklace glomeruli (as in this manuscript), it is (almost) impossible to discriminate between necklace glomeruli and other glomeruli. Consequently, it is very critical to claim that certain glomeruli are necklace glomeruli and consider these glomeruli for any kind of analysis if there is no clear evidence that these glomeruli indeed represent necklace glomeruli. Therefore, I strongly recommend to provide evidence that the glomeruli shown (Fig. 4b) and used for statistical analyses (Fig. 4d) are without any doubt necklace glomeruli. In any other case, these (then) misleading results should be omitted from the manuscript since they would lower the quality. Such dubious analyses with doubtful necklace glomeruli are not required because the authors convincingly demonstrate enhanced expression of the (neuronal) activity marker cFos upon exposure to H₂S in other relevant brain regions (Fig. 4c-d).

Finally, in the legend of Fig. 4b, the sentence "Representative cFOS activity (yellow), mature olfactory sensory neurons (green), nuclei stained with ..." is very problematic. What is shown in this figure is not cFos activity, but cFos immunoreactivity or expression. And there are no olfactory sensory neurons in the olfactory bulb; what is shown in green in Fig. 4b are instead axonal processes of olfactory neurons innervating glomeruli in the bulb.

We thank the reviewer for this important comment. We have now modified the schematic representation of the necklace glomeruli as shown in the revised **Fig. 4a**, indicating that the NG on the OB can be in dorsal, medial, lateral and ventral. Accordingly, we also modified the main text (page 9, line 11 of the new version of the manuscript) and we also added the reference of Bumbalo et al. (2017).

As proposed by the reviewer, we have now provided experimental evidence that the glomeruli on **Fig. 4b** (and the ones used for the statistical analyses) are indeed necklace glomeruli. To show that, we performed immunohistochemistry (on the same slices used for the cFOS antibody staining) with a specific molecular marker for the necklace glomeruli, the Pde1a (Huang, Z., Zimmerman, A. D., & Munger, S. D. *Unique molecular markers for GC-D-expressing olfactory sensory neurons and chemosensory neurons of the Grueneberg ganglion*. bioRxiv preprint (2018)). We have now added a new **Supplementary Fig. 3** with these results. We also adapted the main text accordingly (page 9, line 17-18). The method section on page 20, line 20-23 was also modified adding the antibody used : "[...] To identify the necklace glomeruli in the OB, a second primary antibody Rabbit anti-Pde1a was used (1:50; Thermo Fisher Scientific, MA, USA; PD1A-112AP) on the same tissue slices where the Rabbit anti-cFOS was previously tested [...]".

We thank the reviewer for these comments and have now changed and adapted the figure legend accordingly: "Representative cFOS expression (yellow), axonal processes of olfactory neurons innervating glomeruli in the bulb (green), nuclei stained with DAPI (cyan) showing the neuronal stimulation induced by H₂S". We have also modified **Fig. 4b** where "OMP" has been replaced by "GFP" as requested below.

5) page 10 line 20: *It is claimed that "responses of GG neurons showed no desensitization". Maybe I have overlooked the relevant experiments that substantiate this concept. However, from my point of view, the authors only show that increasing concentrations of H₂S lead to enhanced calcium signals in GG neurons when these cells were exposed to pulses of H₂S (Fig. 2b). In my opinion, to test for desensitization, the cells should be exposed to a given stimulus over a longer period of time or*

repeatedly exposed to the same stimulus (same H₂S concentration) with only rather short breaks between stimulations. Therefore, I suggest to remove the above-mentioned sentence.

We agree with the reviewer, and we have now removed the mentioned sentence from the discussion.

6) *page 16 line 21-23 and Fig. 2d-f: The authors say that "The ratio of the amplitude for each response of the perfusions were measured and divided by the amplitude of the KCl*

perfusion". What is the ratio of the amplitude and how can you measure it? I assume that the authors mean that not the ratio, but the magnitude of the amplitude for each response of the perfusions was measured and divided by the magnitude of the amplitude of the KCl perfusion. Thus, they calculated the ratio. Correct?

If my assumption is true, then it is also doubtful to give the relevant values in a. u. (arbitrary

units?) as indicated in Fig. 2d-f. From a mathematical point of view, dividing a value in a given unit by another value in the same unit results in NO unit and not in an arbitrary one.

Likewise, also for Figs. 3d and 3f, it might be problematic to use the a. u./arbitrary units since I assume that again the magnitude of the amplitude for each response of the perfusions were measured and divided by the magnitude of the amplitude of the perfusion with ATP (it is not specified in the manuscript that the response to ATP was used as a reference value). Correct?

Finally, this criticism may also apply to Fig. 3g as for this figure that is based on experiments

with GG neurons, the authors do not specify in the figure legend what was used as a reference value. The response to KCl?

We would like to thank the reviewer for his/her comment, and we apologize for the confusion in the explanation of the data analysis. The magnitude of the amplitude of each response to the perfusions was indeed measured and divided by the magnitude of the amplitude of the KCl perfusion, for the experiments performed with GG neurons (**Fig. 2**). For the experiments with cells, the same method was used by dividing by the magnitude of the amplitude of the ATP perfusions (**Fig. 3**). We now modified the explanation in the method section (page 17, line 4-5 and page 18 line 22-25) of the new manuscript).

Accordingly, the mention of a.u for the ratio amplitude responses is indeed not correct. Taking this into consideration, we removed a.u. from the graphs of **Fig. 2d-f** and from the legend of the figure as well as from the **new Fig. 3e-f** (previously Fig. 3f-fg) and from its respective legend.

We apologize for the confusion, and we have now tried to describe more clearly the results displayed in Fig. 3g (**Fig. 3f** in the new version of the manuscript) in its legend. Indeed, the experiments shown were performed on HEK cells and not on GG neurons.

7) *Throughout the entire manuscript, there is controversial information about the H₂S concentration used for the experiments. In the legend of Fig. 1 (page 28), it is said that "NaHS (H₂S donor) was used at different concentrations as scale reference (0, 0.2, 1, 5, 25 and 125 μM)". In the Methods section (page 14), however, it is stated that "NaHS*

(Sodium hydrosulfide hydrate; Sigma-Aldrich; Merck, Germany), a donor of H₂S, was diluted in ddH₂O to calibrate the system at different concentrations of H₂S (0, 0.2, 1, 5, 25 and 125 μM)". Therefore, it is unclear whether the above-mentioned concentrations (0, 0.2, 1, 5, 25 and 125 μM) refer to NaHS or to H₂S.

We indeed used NaHS as a H₂S donor to obtain the finally mentioned concentrations of H₂S. We now verified the complete manuscript and whenever concentrations are mentioned, they always correspond to H₂S. We adapted the legend of **Fig. 1** by changing : [...] "NaHS (H₂S donor) was used at different concentrations as scale reference (0, 0.2, 1, 5, 25 and 125 μM)" [...] by the new sentence: "[...] NaHS (H₂S donor) was diluted in ddH₂O to calibrate the system at different concentrations of H₂S (0, 0.2, 1, 5, 25 and 125 μM) (wells A-B) [...]".

8) *Fig 1b and page 28 line 13: "A-C" should read "A + C" and "B-D" should read "B + D" (otherwise it is wrong and confusing)*

We have made the requested change. Fig. 1b and page 31 line 13, the "A-C" and the "B-D" were replaced by "A + C" and "B + D".

9) *Fig. 1e: What is indicated by the numbers on the x-axis? The H₂S or NaHS concentration in μM?*

We thank the reviewer for this comment. As detailed in the point number 7, all the concentrations indicated are the concentrations of H₂S.

10) *Fig. 2d: What do the light grey lines indicate/connect? Data points from the same cell in the presence and absence of Ca²⁺? Unfortunately, this is not mentioned in the figure legend (this also applies to Fig. 2e-f).*

Indeed, the light grey lines indicate the same cell in the presence and or in the absence of Ca²⁺. During the same experiments, we tested the two conditions (**Fig. 2d**). The same experimental approach was used on **Fig. 2e-f** in which we performed also a "rescue".

Accordingly, we now added the light grey lines and the explanation in the legend of Fig. 2: "Grey light lines connect the same neuron for the different tested conditions".

11) *For the calcium imaging experiments with GG neurons, the number of analysed cells is given in the legends of Figs. 2 and 3g. However, there is no information provided in these figure legends, indicating the number of animals (pups) from which these neurons were collected.*

We would like to mention that the experiments described in **Fig. 3g** (corresponding to **Fig. 3f** in the new version of the manuscript) were on transfected HEK cells and not on GG neurons.

We would like to thank the reviewer for his/her comment on the missing mention of the number of pups used. We have now added these numbers to the legend of **Fig. 2**.

- *Fig. 2a* : “[...] Quantification of rpS6 intensity for each neuron after H₂O (control in white; n=46 neurons; number of pups=3) and H₂S (grey; n=59 neurons; number of pups=3) stimulation [...]”
- *Fig. 2d* : “[...] ACSF Ca²⁺ vs. in a ACSF Ca²⁺ free solution for each neuron (SBT, n=23-35; 2-PT, n=6-11; TMT, n=45-52; H₂S, n=42-49 neurons; number of pups=3) [...]”
- *Fig. 2e-f* : “[...] Ratio amplitude responses for each GG neuron for H₂S (25 μM), H₂S-depleted and H₂S (25 μM) (e; n=51; number of pups=3) and for the experiments with the secretions of the skunk, skunk H₂S-depleted and the skunk again (f; n=11; number of pup=1) [...]”

12) *Fig. 2a-b*: In the two micrographs depicted in *Fig. 2a*, OMP-specific staining is indicated. However, it is rather unlikely that this is true. If I am correctly informed, in the genome of the OMP-GFP mice used for this experiment, the OMP-encoding region was replaced by a sequence coding for the green fluorescent protein (GFP) (Potter et al. 2001; *J Neurosci.*; 21:9713-9123.), raising the question how an OMP-specific staining is possible if there is no expression of OMP. I assume that "OMP" in this figure has to be replaced by "GFP". This would be consistent with the relevant figure legend that claims that the green color denotes GFP expression.

The upper panel of *Fig. 2b* presumably also shows a tissue section through the GG of an

OMP-GFP mouse (although this is not clear from the figure legend). If this notion is true, the respective figure legend (page 30 line 7-8) "Microscopy images of a tissue slice of a mouse GG observed with a DIC contrast (Hv) showing neurons in green (OMP) and after Fura-2 loading" should be better replaced by "Microscopy images of a tissue slice through the GG of an OMP-GFP mouse observed with a DIC contrast (Hv) showing GG neurons in green (GFP) and after loading with Fura-2".

The authors thank the reviewer for this comment. We now have changed, in the **Fig. 2a-b**, "OMP" by "GFP". We also modified the legend of the **Fig. 2b** accordingly. We also applied this change to the entire manuscript.

13) page 34 line 11 and *Fig 4d*: What means "quantification of the density of cFOS+ cells"?

In *Fig. 4d*, there are values indicated, but no unit is provided for this density. In the Methods section (page 19 line 11), the authors only speak about the number of cells. Please specify.

For the quantification of cFOS-positive cells. In the OB, more precisely the necklace glomeruli, the quantification was based on the number of cFOS+ cells considering a radius of 60 μm, as a criterion in imageJ software, around each reconstituted glomerulus.

For the other brain regions (La, VMHDM and PAG), a zone of 350 x 350 μm was used for the counting of the number of cFOS-positive cells using imageJ software.

We modified the method section page 21, line 4-9 of the new manuscript accordingly as well as the legend of *Fig. 4*. We also changed **Fig. 4d** replacing "Density" by "Number of cFOS+ cells" on the Y axis of the graphs.

14) The headline of the legend of *Supplementary Fig. 1* ("Axo mice have no main odorant

detection deficit ...") requires revision. What do the authors mean with main odorant detection and what is a main odorant? Do the authors mean detection of odorants via the main olfactory system? This heading is puzzling. I therefore suggest replacing "Axo mice have no main odorant detection deficit ..." with "Axo mice have no deficit in food detection ...", as they are obviously not impaired in their search for food (a cookie).

As suggested by the reviewer, we made the change in the **Supplementary Fig. 4** (formerly Supplementary Fig. 1): "Axo mice have no main odorant detection deficit ..." with "Axo mice have no deficit in food detection ...".

Minor aspects

15) For some experiments, sulfidefluor-7 acetoxymethyl ester (SF7-AM) is used as H₂S probe. What does that mean? Does SF7-AM bind to H₂S and does it change its fluorescence upon binding? It would be helpful for the reader to provide some relevant information about SF7-AM in the text (e.g. Methods section) and not only referring to a reference.

The authors thank the reviewer for this important comment. We now added an explicative sentence for the mechanism of action of SF7-AM (page 14 line 12-14): "The azide moieties of SF7-AM will react with H₂S to generate carboxamide rhodamine 110, which has wavelengths of 488 nm (excitation) and 535 nm (emission)".

16) At several places throughout the manuscript (e.g. page 5 line 8) the authors use the term "animal secretions". What do they mean exactly? Urine? Please specify.

We now added more precisions about the animal secretions. On page 5 line 11-12, we added the sentence: "Except for the skunk where anal gland scents were used, for the other animals, we used urine". We also modified on page 15 line 4-5, in the method section, *Spectrofluorimetry*, adding the sentence in parenthesis: "[...] to quantify the concentration of H₂S in predator secretions (anal gland scents for the skunk and urines for the others) [...]".

17) The term "fisher" is used (page 5 line 9 and legend of Fig. 1) What does that mean? A (human) fisherman? In this context, for all the mammalian species employed in this study (mouse, elk, cattle, rabbit, raccoon, fox, fisher, mountain lion, and skunk) the correct scientific name(s) should be given as well (e.g. *Puma concolor* for the mountain lion) to avoid such misunderstandings.

The authors thank the reviewer for this comment. We now added the latin name of each mammalian species on page 5, line 8-11: "[...] non-predators: elk (*Cervus canadensis*), cattle (*Bos taurus*) and rabbit (*Oryctolagus cuniculus*) and from predators: raccoon (*Procyon lotor*), fox (*Vulpes vulpes*), fisher (*Pekania pennanti*), mountain lion (*Puma concolor*) and skunk (*Mephitis mephitis*) [...]". We added them also in the legend of **Fig.1**.

18) page 6 line 5: What is GFP? Green fluorescent protein? Please explain this abbreviation.

We now changed the sentence page 6, line 7-8: “[...] where the reporter GFP [...]” to “[...] where the green fluorescent protein (GFP) [...]”.

19) page 6 lines 5-6: *It is claimed that expression of GFP under the control of the promoter of the olfactory marker protein (OMP) allows the visualization of “all mature olfactory sensory GG neurons”. Has it ever been shown that the OMP promoter is active/activated in ALL mature GG neurons? Therefore, I recommend omitting the word “all”.*

We now changed the sentence page 6, line 9 omitting the word “all” as recommended by the reviewer.

20) page 7 line 6: *The sentence “These results demonstrated that H2S acts as a kairomone via the activation of GG neurons ... ” is not correct - at least not at this text passage. Up to this passage in the text, it has been only shown that H2S is released by predators and activates GG neurons. Evidence that H2S actually acts as a kairomone by triggering certain physiological and behavioral responses is provided later in the text. Therefore, the relevant sentence should read “These results demonstrated that H2S activates GG neurons”.*

We have now changed the sentence page 7, line 11 according to the reviewer’s request.

21) page 7 lines 14-15: *At this, but also at other text passages, the wording regarding proteins that are expressed in human cells using plasmid vectors should be improved. Consequently, the sentence “... in which we transfected a reporter plasmid, GFP and the murine CNGA3 plasmid” should better read “... in which we transfected a reporter plasmid encoding GFP and a second plasmid coding for the murine CNGA3 channel”. At least, I assume that the authors did not transfect cells with a reporter plasmid, GFP, and the murine CNGA3 plasmid (what is a murine plasmid?) but with two plasmids of which one encodes GFP and the other the murine CNGA3 channel. Please consider this aspect also for other relevant passages of the manuscript (such as the legend of Fig. 3).*

We thank the reviewer for this useful comment. We now changed the sentence page 7, lines 19-20: “[...] in which we transfected a reporter plasmid, GFP and the murine CNGA3 plasmid [...]” to “[...] in which we transfected a reporter plasmid encoding for GFP and a second plasmid coding for the murine CNGA3 plasmid [...]”. The legend of **Fig.3** was also changed accordingly.

22) *The description of cells lacking or being endowed with CNGA3 or cFos should be improved (e.g. page 7 line 19, page 8 line 11, legend of Fig. 3). For instance, in the legend of Fig. 3, a “CNGA3 – cell” is mentioned. It is not immediately clear whether this cell expresses or lacks CNGA3. Therefore, it would be better to designate “- cells” as CNGA3-negative (or cFos-negative) and “+ cells” as CNGA3-positive/expressing (or cFOS-positive/expressing) cells (throughout the entire manuscript).*

We changed in the main text “CNGA3 –” by “CNGA3-negative” and “CNGA3 +” by “CNGA3-positive”. We also clarified it in the legend of **Fig.3**. We did the same for “cFOS+” by replacing it by “cFOS-positive”.

23) page 8 line 14: The sentence " ... implicated in the treatment of danger information ..." should read "... implicated in the processing of danger information ..."

We changed the sentence page 9, line 19: "[...] implicated in the treatment of danger information [...]" with the proposed sentence: "[...] implicated in the processing of danger information [...]".

24) page 10 line 4: In the sentence "... behaviors in the preys such as the mice" should read "... behaviors in mice" since we do not know whether H2S also elicits similar responses in other prey animals, do we?

We changed the sentence page 11, line 8: "[...] behaviors in the preys such as the mice [...]" to "[...] behaviors in mice [...]".

25) page 11 line 11-12: The sentence "Thus, H2S present in the predator secretions could induce in the prey this artificial hibernation as a survival strategy" is very puzzling. Has hibernation ever been observed in *Mus musculus*? Presumably, it is better to omit this sentence from the manuscript.

We thank the reviewer for this comment. As we mentioned in the paper, H2S at very high concentrations induces a suspended animation-like state in mice (Blackstone et al., 2005). This specific behavior can be compared to an "artificial hibernation" that can be induced by an innate fear molecule, 2-methyl-2-thiazoline (2MT) (Matsuo et al., 2021). In the wild, in extreme cold, mice enter a similar torpor – a hibernation-like state of slowed metabolic rate and reduced body temperature (Hrvatin et al., 2020).

26) page 14: The authors describe the spectrofluorimetry approach used to quantify the H2S concentration in several solutions and predator secretions. In the description of this experiment, it was not clear to me what they used the fluorescence stereomicroscope (Leica M165 FC) for because visualization of the H2S concentration is not shown (at least not in micrographs). Moreover, in this paragraph, the authors should explain the abbreviations ddH2O (double distilled water?) and RT (room temperature?).

Fig. 1b shows, on its right panel, the fluorescence of SF7-AM in the presence of H2S with real acquisitions of the ELISA plate wells obtained with the fluorescence stereomicroscope (Leica M165 FC). These acquired micrographs have been added to a schematic representation of the plastic plate. This is now described in the legend of **Fig.1b** of the revised manuscript.

ddH2O indeed means double distilled water and RT, room temperature. We have now changed these abbreviations to "water" and "room temperature" to avoid confusion.

27) page 15 line 9: For the OMP-GFP mice, the reader is referred to reference 30. As far as I know, this is not the original publication in which the generation of this transgenic mouse line has been initially described and characterized. Thus, it would be desirable that the reader is referred to the original publication.

We thank the reviewer for pointing this out to us. Indeed, it was a mistake in the referencing method. We changed the reference to Potter et al., 2001 (Reference n° 32 now; reference n°31 in the previous version of the manuscript).

28) *With respect to the latter aspect, it is a flaw of the present manuscript that in particular regarding the GG, its molecular phenotype, and its ability to detect alerting compounds, the reader is only referred to a few publications that are mostly from the authors' own group. For instance, concerning the localization of CNGA3 in cilia of GG neurons, the reader is referred to reference 32 (a publication of the authors' own group). However, this observation has been initially published several years before by another group (Liu CY, Fraser SE, Koos DS. Grueneberg ganglion olfactory subsystem employs a cGMP signaling pathway. J Comp Neurol 516:36-48,2009). Thus, it seems that the authors are only aware of their own publications. In my opinion, such self-citation and neglect of the work of others does not meet the high scientific standards given by this journal. Authors should therefore consider to cite the work of others at appropriate passages in the text.*

We apologize for the missing reference concerning the localization of CNGA3 in the cilia of GG neurons. We now mentioned the Liu et al (2009), reference n°36 in the new version of the manuscript. Accordingly, we made sure to reference appropriately the entire manuscript.

29) *page 16 line 4 and page17 line 24: the sentence "observations were performed at 480 nm (GFP) and 380 nm (Fura-2) ..." is not very precise. I assume that the given wavelengths (480 nm for GFP and 380 nm for Fura-2) denote the excitation wavelengths, don't they? What about the emission wavelength(s)? Please specify.*

Indeed, we only mentioned the excitation wavelengths. To be more precise, we now have changed in the Methods part, section *Calcium imaging experiments on GG tissue slices* (page 16 line 16-17) as well as in the section *In vitro expression of CNGA3 channels* (page 18 line 13-14): "Observations were performed at ex 480 nm / em 508 nm (GFP) and ex from 380 nm (Ca²⁺-free) to 340 nm (Ca²⁺-saturated) / em 510 nm (Fura-2)...".

30) *page 18 line 1 : "Adenosine 5'-triphosphate in Ringer solution (100 µM; Sigma-Aldrich ® ATP; Merck, Schafftrausen, Switzerland) was perfused ..." should read "Adenosine 5'-triphosphate (ATP; 100 µM; Sigma-Aldrich Merck, Schaffhausen, Switzerland) in Ringer solution was perfused ...".*

We changed the sentence page 18, line 15: "[...] ATP; 100 µM; Sigma-Aldrich® Merck, Schaffhausen, Switzerland [...]".

31) *page 18 line 19: Which buffer was used to dissolve the 4% PAF? How long did the fixation step take?*

The paraformaldehyde solution (PAF) was prepared in a phosphate-buffered saline with a pH of 7.6 to obtain a 4% PAF solution. The fixation step duration was 12 hours (overnight). We now added these experimental precisions on page 20 line 10-11.

32) page 19 line 14: *How long were the mice exposed to H2S?*

The duration of H2S stimulation is 5 minutes as indicated on page 21, line 13: "Briefly, mice were placed in contact with 200 μ L of H2O or H2S on a blotting paper for **5 min** in the home cage."

33) page 20 line 6-7: *What means "excessive animal movements"?*

We refer to "excessive animal movements," when the mice exhibited strong escape behaviors despite undergoing a 5-day habituation period to the procedure. This includes behaviors such as vigorous struggling or continuous body twisting, which can interfere with accurate blood pressure measurements. Since proper habituation typically results in calm and stable positioning of the animals, excessive movements suggest that the individual mouse was either not sufficiently habituated or experiencing an external stressor that prevented reliable measurements.

Accordingly, we now defined these excessive animal movements in the Method section: "Measurements obtained in the presence of excessive animal movements (i.e., vigorous struggling or body twisting, indicating insufficient habituation or external stress) were discarded".

34) page 20 line 18: *When did the recording phase of 5 min start? Immediately after placing the blotting paper treated with H2O or H2S?*

The blotting papers were added first in the open field arena. Then, the test session started immediately when the mice were introduced in the open field. We added a sentence with more precision on page 22 line 18-19: "The recording session begun once the mice were introduced in the arena containing the blotting papers."

35) page 21 line 13-14: *"... GG neurons was detected thanks to their OMP expression" should read "... GG neurons was detected thanks to their GFP expression"*

We changed the sentence on page 23, line 16-17: "[...] thanks to their OMP expression [...]" to "[...] thanks to their GFP expression [...]".

36) page 28 line 7: *It would be better to replace "cut" by "section"*

We changed the sentence on page 31, line 7 by replacing the word "cut" by the word "section".

37) Fig. 3b: *"24h 48" should read "24-48 h"*

We modified **Fig. 3a** (formerly Fig. 3b) changing "24h-48" to "24-48h".

38) page 32 line 16: *I recommend to replace "percentage of responding cells (left)" by "percentage of responding HEK cells (left)" to avoid any confusion with GG cells for which the results are shown in the next figure (Fig. 3g).*

We apologize for the misunderstanding of **Fig. 3**. All the data shown in this figure were obtained on HEK cells (non-transfected or transfected cells with CNGA3). As mentioned on **question 6**, we adapted the legend of the figure to avoid any confusion.

39) *Fig. 5c: What means BPM? Beats per minute?*

We now mentioned the meaning of BPM (beats per minute) in the legend of **Fig. 5c**.

40) *Figs. 5e and 5g and Supplemental Fig. 1b: "Freezing" should be placed by "Time of freezing".*

Moreover, in these figures, the presentation of the data regarding the standardized index is somewhat unusual and its calculation not entirely clear (since it has not been described in the manuscript). Let's assume that intact control mice exposed to H₂O have covered a total distance of 1 meter (on average) within the period of observation (Fig. 5e). For comparison with other intact mice exposed to H₂S, wouldn't it make sense to set this (average) total distance of 1 meter at 100% and not 0% (as the authors apparently did)?

For Fig. 5g, it is unclear which mouse was compared with which mouse. I assume that Ctrl

mice exposed to H₂O were compared to Ctrl mice exposed to H₂S. And Axo mice exposed to H₂O were compared to Axo mice exposed to H₂S. Furthermore, I assume that the values for the Ctrl mice in Fig. 5g were taken from the experiment(s) depicted in Fig. 5e. If this assumption is correct, then the authors should describe it like that in the corresponding figure legend. Their description "and by the comparison of the different parameters (g) previously measured in (e)" is difficult to understand for the reader because it is unclear what comparison of the different parameters means.

In Supplementary Fig. 1b, it is not clear which value was set as the reference (i.e. 0%).

I

suspect that Ctrl mice exposed to H₂O were used as a reference for comparison with the mice in the other experimental conditions. Therefore, the authors should clearly describe this in the figure legend.

We apologize for the lack of clarity of the data representation. On **Fig. 5e (and g)** and in the **Supplemental Fig. 4b** (previously, **Supplemental Fig. 1b**), we modified "Freezing" by "Time of freezing".

Regarding the calculation of the standardized index, if we use your example: Control mice were first exposed to H₂O vs. H₂O in what we called the control session, during which they covered a total distance of approximately 1 meter. The test session was performed 2–3 hours later. In this session, control mice were again exposed to H₂O vs. H₂O, while another group of control mice was exposed to H₂O vs. H₂S.

For the mice tested with H₂O vs. H₂O, in both sessions, their total distance covered remained similar between sessions (i.e., around 1 meter in each session). As a result, the difference between the test session and the control session was close to 0, and we standardized this difference to 0% (represented as white bars in **Fig. 5e**). For the mice exposed to H₂O vs. H₂S, in the test session, their total distance covered was reduced compared to their own control session (H₂O vs. H₂O). This decrease resulted in a negative index, as shown by the grey bars in **Fig. 5e**.

To be more precise, we now modified the description of the method on page 23 line 1-5, "Test sessions (H₂O vs. H₂O / H₂O vs. H₂S, 25 or 125 μ M) were performed

at least 2–3 hours after the control sessions (H2O vs. H2O). The difference in mean distance travelled between the test session and the control session for the H2O vs. H2O group was set at 0%. The difference in mean distance travelled between the test session and the control session for the H2O vs. H2S groups was standardized using the H2O group as a reference.”

For **Fig. 5g**, Ctrl mice and Axo mice are compared for the condition with H2S. So again, control session was H2O vs. H2O and the test session H2O vs. H2S for Ctrl and Axo mice. The results for the Ctrl mice are the ones obtained on Fig. 5e. We added more precision in the legend of Fig. 5g: “[...] and by the comparison of the different parameters (g) between Ctrl mice (results taken from (e) for H2S condition) and Axo mice in the presence of H2S.”

For the **Supplemental Fig. 4b** (formerly Supplemental Fig. 1b), as mentioned above, the references set at 0% is the condition H2O (test session (H2O vs. H2O) – control session (H2O vs. H2O) for Ctrl mice and Axo mice.

Hydrogen sulfide as a potent predator-derived kairomone mediating fear-related responses in mice

Responses to the reviewers:

Reviewer #1

General remarks

The effort of the authors to perform necessary, additional experiments has improved the manuscript substantially. I have only few comments on the revised manuscript.

We would like to thank the reviewer for his/her very appreciated comments and for the new questions/remarks raised. Please, find here our point-by-point responses.

Comments

1. *Fig. S2b, d Use the same font size for x-axis labeling (preferably the larger font!)*

We thank the reviewer for this comment. We have now changed the x-axis labeling of Fig. S2b to a larger font.

2. *Fig. S2b,d About 20% responding cells at 200 mM IBMX makes sense, and reasonably depicts a transfection efficiency. In this context it is not clear what '100% responding cells' in Fig. S2d, left panel mean. Does it mean that all cells that responded to H2S in IBMX-Ringer also responded to H2S in Ringer without H2S? Please clarify.*

We thank the reviewer for this question. The presence of 100 μ M IBMX did not affect the observed effect of H2S. Indeed, in Fig. S2d, all the cells responding to H2S + IBMX at 100 μ M also respond to H2S in Ringer solution (without the inhibitor, IBMX). We had previously tested a gradient of IBMX concentrations on transfected cells (Fig. S2b) and decided to use 100 μ M IBMX with H2S as, at this IBMX concentration, the cells did not show any response to IBMX. To be more precise, we have now decided to change the main text accordingly (Results section, page 8 line 3-10) :

“To ensure that the responses observed with H2S are due to a direct activation of CNGA3 and not to an indirect increase in cyclic nucleotides by a potential inhibition of phosphodiesterase (PDE)⁷, we used perfusions of an IBMX gradient (**from 2 μ M to 200 μ M**) on CNGA3-positive cells. In the presence of this PDE inhibitor, we did not observe any increase in the intracellular calcium **up to 100 μ M** of IBMX, **a concentration known to efficiently inhibit PDE⁷** (Supplementary Fig. 2a-b). We then tested the activation of the CNGA3 channel by H2S in the presence of IBMX, **at this 100 μ M concentration** and observed no significant differences in the responses with or without the inhibitor [...]”.

3. Fig. S2a, c The vertical scale bars in panels a, c should be labelled with ratio numbers, as done in b, c (right panels). I suppose there are no units ([au]) since this is a FURA-2 ratio of 515 nm emission signals induced by stimulation with either 340 or 380 nm...

We thank the reviewer and apologize for the lack of clarity. Fura-2 AM is a ratiometric fluorescent calcium indicator that will bind free intracellular calcium ions. Fura-2 is excited at two different wavelengths depending on its binding with calcium: 340 nm excites the calcium-bound form and 380 nm excites the calcium-free form. As you mentioned, there is no unit for the FURA-2 for Fig. S2a, c (Ratio=Fura340/Fura380). For all calcium imaging experiments, the same parameters were always applied. To be more precise, we changed the sentence in the method section on page 16, line 18-20 and on page 18, line 18-20:

“Observations were performed with the 25x objective at ex 480 nm / em 508 nm (GFP) and ex from 380 nm (Ca²⁺-free) to 340 nm (Ca²⁺-saturated) for the Fura-2 as it is a ratiometric fluorescent calcium indicator with an emission at 510 nm.”

When we mentioned the ratio amplitude of the responses on Fig. S2b, d (right panels), we are indeed talking about the measured amplitude of the cues divided by the measured amplitude of the ATP response.

Hydrogen sulfide as a potent predator-derived kairomone mediating fear-related responses in mice #COMMSBIO-24-4865C

Responses to the reviewers:

Reviewer #2

General remarks

The manuscript by Lopes and co-workers was significantly improved by the changes made by the authors. In particular, the authors have addressed many of the critical points raised by the reviewers. Therefore, I can endorse the publication of this manuscript. However, there are still some major and minor aspects that require additional attention and should be clarified before publication.

We would like to thank the reviewer for his/her very appreciated comments and for the new major and minor aspects raised. Please, find here our point-by-point responses.

Major aspects

Q1a. *page 9 line 14-18: The sentences “We observed a significant increase in the number of cFOS-positive nuclei after stimulation with H₂S (+69 % of activation) compared to the baseline activation observed with H₂O stimulation in one represented glomeruli (Fig. 4b and d). cFOS+ NG were also systematically tested for the phosphodiesterase 1a (Pde1a) expression, a specific marker of the NG45 (Supplementary Fig. 3).” are difficult to understand. Was only one glomerulus actually analyzed? Probably not. Thus, I assume, it should read “Compared to the baseline activation observed with H₂O stimulation, we observed a significant increase in the number of cFOS-positive cells (+69 %) surrounding glomeruli after stimulation of animals with H₂S (Fig. 4b and d). For this analysis, only glomeruli were considered that were immunoreactive for phosphodiesterase 1a (Pde1a), a specific marker of the NG45 (Supplementary Fig. 3).” Is that what the authors wanted to convey? Furthermore, the figure of 69 % cannot be correct because the bar for H₂S is almost three times as large as that for H₂O. Consequently, there should be an increase of about 200%.*

R1a. We thank the reviewer for this comment. We had previously mentioned a 69% difference and we have now replaced this information by the % of increase/activation as required by the reviewer. We have now reformulated accordingly on page 9, line 14-18:

“Compared to the baseline activation observed with H₂O stimulation, we observed a significant increase in the number of cFOS-positive cells (+**220** % of activation) surrounding glomeruli after stimulation of animals with H₂S (Fig. 4b and d). For this analysis, we considered only the glomeruli that were immunoreactive for phosphodiesterase 1a (Pde1a), a specific marker of the NG⁴⁵ (Supplementary Fig. 3).”

We decided to change also this indication (difference to activation) for the VMHDM and the PAG zones (page 10, line 3-4) as well.

Q1b. *Importantly, another aspect remains puzzling: the authors claim in their rebuttal letter that for the revised version of the manuscript, they have only analyzed glomeruli positive for the necklace glomeruli (NG) marker Pde1a. This does not appear to have been the case for the initial version of the manuscript in which cFos-expressing cells were counted regardless whether the relevant glomerulus was Pde1a-positive or Pde1a-negative (at least a co-staining with an antibody for Pde1a was not mentioned in the original version). In this context, it is very surprising that the data set (i.e. the numbers in the bar chart in Fig. 4d) presented for this experiment appears to be the same as in the original version of the manuscript.*

R1b. For the initial version of the manuscript, we had already verified that all glomeruli analysed and presented in Fig. 4d expressed Pde1a, through co-stainings. While these co-stainings with a Pde1a antibody were not explicitly mentioned in the first version, we can confirm that we only quantified cFos-positive cells associated with Pde1a-positive glomeruli, consistent with our focus on necklace glomeruli. As a result, the dataset in Fig. 4d remains unchanged.

2. page 16 line 16-17 and page 18 line 13-14: It seems that the excitation wavelength used for calcium imaging experiments varies between calcium-free (380 nm) and calcium-saturated (340 nm) conditions. Why is this so? Presumably, this could influence the experimental results. Can these results still be compared (for instance in Fig. 2c) if the excitation wavelength was not identical?

We thank the reviewer, and apologize for the lack of clarity. Fura-2 AM is a ratiometric fluorescent calcium indicator that will bind free intracellular calcium ions. Fura-2 is excited at two different wavelengths depending on its binding with calcium: 340 nm excites the calcium-bound form and 380 nm excites the calcium-free form. As you mentioned, there is no unit for the FURA-2 for Fig. S2a, c (Ratio=Fura340/Fura380). For all calcium imaging experiments, the same parameters were always applied. To be more precise, we changed the sentence in the method section on page 16, line 18-20 and on page 18, line 18-20:

“Observations were performed with the 25x objective at ex 480 nm / em 508 nm (GFP) and ex from 380 nm (Ca²⁺-free) to 340 nm (Ca²⁺-saturated) for the Fura-2 as it is a ratiometric fluorescent calcium indicator with an emission at 510 nm.”

3. Page 34 line 8: Were the 11 neurons analyzed in Fig. 2f actually from only one pup? Is that representative?

We thank the reviewer for this remark. The purpose of the experiment was to test, at a cellular level, the impact of the lack of H2S in a biological sample. For that, we used one pup that gave us 3 GG slices and 11 neurons. As this **control** experiment across slices/cells was highly reproducible and performed “paired-way” (Skunk vs. Skunk H2S-depleted), we considered that this one control experiment support our conclusions.

4. Fig. 5e + 5g + Suppl. Fig 4b (including the corresponding figure legends and the description of these results in the methods section): The calculation of the “index standardized [%]” is still absolutely elusive. The authors say in their rebuttal letter that “Control mice were first exposed to H2O vs. H2O in what we called the control session, during which they covered a total distance of approximately 1 meter. The test session was performed 2–3 hours later. In this session, control mice were again exposed to H2O vs. H2O, while another group of control mice was exposed to H2O vs. H2S. For the mice tested with H2O vs. H2O, in both sessions, their total distance covered remained similar between sessions (i.e., around 1 meter in each session). As a result, the difference between the test session and the control session was close to 0, and we standardized this difference to 0% (represented as white bars in Fig. 5e).” Consequently, if in a H2O vs. H2O approach, the difference between the test session and the control session was close to 0 meter, and the authors standardized this difference to 0%, this would mean that a difference of 0 meter was defined as 0%. Right? For the mice exposed to H2O vs. H2S, in the test session, the authors claim that “their total distance covered was reduced compared to their own control session (H2O vs. H2O). This decrease resulted in a negative index, as shown by the grey bars in Fig. 5e.” Let us assume that these animals also covered 1 m under control conditions (H2O versus H2O) but only 0.6 meter under test conditions (H2O versus H2S). In this scenario, the difference between the test and the control session would be -0.4 meter. If 0 meter correspond to 0% (as mentioned by the authors), what would be the corresponding percentage for -0.4 meter? From my point of view, this type of calculation does not make sense. Therefore, the authors should provide a clear, comprehensive and mathematically correct formula for calculating this so-called “index standardized [%]”. Such a formula should not only be given for the total distance, but also for entries central zone, distance central zone, entries danger zone and time of freezing.

We thank the reviewer for this comment. For each behavioral parameter, a relative index represented in percent, according to Brechbühl et al. 2015, was calculated in two steps.

First, for each mouse and each condition (H2O and H2S), we subtracted the value obtained during the control session from the value obtained during the test session, resulting in an individual difference. Then, the mean of these differences in the H2O condition (the reference) was subtracted from all values (H2O and H2S), in order to center the H2O distribution at 0% and to express the effect of H2S relative to this baseline.

The formula used was: $\text{Index}_i = (X_{\text{test},i} - X_{\text{control},i}) - \mu_{\text{H}_2\text{O}}$

To be more precise, we modified the method section accordingly on page 23, line 5-10:

“For each behavioral parameter, a relative index represented in percent was calculated in two steps²³, using the formula: $\text{Index}_i = (X_{\text{test},i} - X_{\text{control},i}) - \mu_{\text{H}_2\text{O}}$. Briefly, for each mouse and each condition (H2O and H2S), we subtracted the value obtained during the control session from the value obtained during the test session, resulting in an individual difference. Then, the mean of these differences in the H2O condition was subtracted from all values (H2O and H2S), in order to center the H2O distribution at 0% and to express the effect of H2S relative to this baseline.”

Minor aspects

5. page 7 line 1: “2-PT, -100 %; TMT, 45 % and H₂S, -24 %” should read “2-PT, -100 %; TMT, -45 % and H₂S, -24 %” (the minus sign in relation to 45% should not be missed).

We apologize for this mistake, we have now corrected it.

6. page 11 line 23-25: *It is not at all surprising that increased concentrations of the stimulus (H₂S) lead to enhanced behavioral responses. Initially, this has nothing to do with a lack of desensitization. Therefore, in the sentence “Interestingly, in behavioural experiments, increasing the concentration of H₂S induced accentuated fear-related responses suggesting a lack of desensitization.” should read “In behavioural experiments, increasing the concentration of H₂S also induced accentuated fear-related responses.”*

We have now changed the sentence as proposed by the reviewer.

7. with respect to point 1 of my initial referee report (page 15 line 13-15 of the present manuscript): *The authors describe that to deplete the stock solution of H₂S, they added 500 µl of 5 M HCl to a stock solution, which contained 2 mM of NaHS, corresponding then to 500 µM of H₂S. Unfortunately, the volume of the NaHS stock solution to which 500 µl of 5 M HCl was added is not mentioned. Was it 10 ml or 100 ml or how much of NaHS solution?*

We apologize for the mistake. The volume of the NaHS (2 mM) solution was 5 ml. We now added that volume on page 15 line 14.

8. page 18 line 15: *As already mentioned in my last referee report “Adenosine 5'-triphosphate in Ringer solution (100 µM; Sigma-Aldrich ® ATP; Merck, Schafftrausen, Switzerland) was perfused ...” should read “Adenosine 5'-triphosphate (ATP; 100 µM; Sigma-Aldrich Merck, Schaffhausen, Switzerland) in Ringer solution was perfused ...”.*

We changed the mention of the product details in the sentence.

9. page 19 line 3: *The “the plasmid CNGA3” should read “the plasmid encoding CNGA3”.*

We have made the requested change on page 19 line 9.

10. legend of Fig. 1b (page 31 line 15-16): *The authors claim that micrographs acquired with a fluorescence microscope were added to a schematic representation of the plastic plate. What do the authors mean with “micrographs”? These dots in different shades of green on the right-hand panel of Fig. 1b? By definition, a micrograph is an image, captured photographically (or digitally), taken through a microscope to show a magnified image of an object. I'm sorry, but I can't recognize a micrograph in Fig. 1b. But maybe, I missed it.*

We would like to confirm here that these “dots shown in various shades of green” are indeed micrographs of the wells of the plate captured photographically under the stereomicroscope.

Please find below the images from which each zone was obtained. Here, we can see the H₂S gradient without and with SF7-AM (black rectangle). The red circle is an example of the representative zone of the micrograph that was selected for Fig. 1b.

11. coming back to point 9 of my initial referee report: The description of the x-axis in Fig. 1e (page 31) is still confusing. In their rebuttal letter, the authors claim that all concentrations given are the concentrations of H₂S. This would mean that both the values on the x-axis and on the y-axis would denote the H₂S concentration. Does that make sense? In my opinion, no. If I understood the experiment correctly, the authors determined the H₂S concentrations of NaHS solutions. Wouldn't it therefore make more sense to show the NaHS concentration of these solutions on the x-axis and the resulting H₂S concentration of the respective solution on the y-axis?

We thank the reviewer for this pertinent remark. We have now mentioned the concentrations of NaHS in the x-axis. We also adapted the legend accordingly.

12. legend of Suppl. Fig. 1: “CNGA3 channels are expressed at the cilia ...” should read “CNGA3 channels are located in the cilia ...”.

We have made the requested change.

13. both bar charts in Suppl. Fig. 2d and the corresponding figure legend (line 27): Wouldn't it make more sense to replace “H2S-IBMX” by “H2S + IBMX”, as this solution contains H2S plus IBMX (obviously, it is not H2S minus IBMX)?

We have now made this modification on the Suppl. Fig. 2d and in the corresponding figure legend.

Hydrogen sulfide as a potent predator-derived kairomone mediating fear-related responses in mice

Response to the reviewer:

Reviewer #2

General remarks

The authors have (for the most part) adequately addressed the points of criticism raised by this reviewer. However, there is still one aspect that urgently needs clarification. Otherwise, I can recommend the manuscript for publication in this journal.

We thank the reviewer for recommending the publication of our manuscript. Please, find here our response for the last point of clarification.

Major aspects

1. With respect to point 4 of my last referee report, for Fig. 5e + 5g + Suppl. Fig 4b (including the corresponding figure legends and the description of these results in the methods section), the calculation of the “index standardized [%]” is still absolutely elusive. The authors present the formula: $Index_i = (X_{test,i} - X_{control,i}) - \mu_{H_2O}$. These abbreviations are totally unclear and should be explained. What does μ_{H_2O} stand for, for example? And how does such a formula lead to a value given in %? In addition, the description of the experimental procedure of the relevant behavioral experiments in the methods section is extremely sparse and the literature references provided are not very helpful. In this context, the authors wrote in their recent rebuttal letter that “Control mice were first exposed to H₂O vs. H₂O in what we called the control session.(...). The test session was performed 2–3 hours later. In this session, mice were again exposed to H₂O vs. H₂O, while another group of mice was exposed to H₂O vs. H₂S. This important information is only given very briefly in the methods section. It is therefore very difficult for the reader to understand the experimental set-up. I suggest describing the (entire) procedure of the behavioral experiments in much more detail; in a way that is also clearly comprehensible for the reader.

We thank the reviewer for her/his helpful comment and the opportunity to clarify the calculation and rationale behind the “standardized index [%]” presented in Figures 5e, 5g and Supplementary Fig. 4b.

Index calculation: To facilitate comparison across animals and experimental conditions, we applied a two-step normalization and centering procedure to the behavioral data (e.g., total distance, number of zone entries, freezing duration):

Step 1: Normalization per mouse.

For each behavioral parameter and for each individual mouse, we first expressed the test session value relative to the control session value. Specifically, the control session (H₂O vs. H₂O) value was set to 100%, and the test session value (either H₂O vs. H₂O for the control group, or H₂O vs. H₂S for the test group) was expressed proportionally. This step controls for inter-individual variability and differences in the different parameter units (m, s, number of entries,...).

Step 2: Centering to H₂O mean.

We then computed an index for each individual mouse. This difference index was then calculated for each individual by subtracting the normalized control value (100%) from the normalized test value, yielding a relative change in percentage.

The formula used was: $Index_i = (X_{test,i} - X_{control,i}) - \mu_{H_2O}$

Where:

- $X_{test,i}$ is the value for the test session (in %),
- $X_{control,i}$ is the value from the control session (set to 100%),
- μ_{H_2O} is the mean of all individual differences in the H₂O vs. H₂O (control) group.

This second step corresponds to a centering of the data, so that the mean of the H₂O control group is set to 0%. This allows the distribution of values to be visualized relative to a common, biologically neutral reference which improves interpretability by allowing direct visual comparison of the effect of the test odor (H₂S) relative to the control condition.

Negative index values for distance-related parameters and number of entries indicate increased fear-related behaviors, such as reduced exploration. Positive index values for the freezing parameter reflect fear responses, as freezing is a well-established indicator of fear.

We agree that the original Methods section lacked sufficient details, and we have now revised it. According to the reviewer request, we have detailed better the experimental procedure in the Methods section and quoted an additional relevant reference (reference number 46; page 22, line 18-24). Here is the new description of the entire behavioral procedure (page 22-23) integrating also the above calculation details and the explanation for the abbreviations (page 23, line 10-23):

“These experiments were performed according to Lopes et al., 2022¹⁵. Briefly, an open field test was used to detect and observe fear-related behaviors of mice in the presence of the H₂S (25 or 125 μ M). Mice were housed in the mouse facility with a 12:12 h light/dark inverted cycle in their home cage with food and water ad libitum.

One week before the beginning of the experiments, the animals were familiarized with the test arena to minimize the environmental stress. The test arena consisted of a closed Plexiglas box (45 × 25 × 19 cm) and 2 pieces of blotting paper (3 × 3 cm) in the middle of the width of the arena. The arena was divided in three different zones⁴⁶: a safety zone, a central zone and a danger zone. A piece of blotting paper was positioned in the safety zone with a systematic addition of 1 ml of H₂O and another blotting paper was added in the danger zone with 1 ml of the tested cue (H₂O or H₂S at different concentrations). The presence of a blotting paper on both zones avoid any visual influence and focuses the test on the olfactory effect of the cues. In this arena, comparisons can be made between safety zone (H₂O) vs. danger zone (H₂O or H₂S). Blotting paper of the size of the box covered the floor to avoid direct contact of the mice with the Plexiglas.

Mice behaviors were recorded for at least 5 min during the nocturnal phase from the top of the arena covered with a Plexiglas plate by IR-sensitive HD camera under infrared vision. The recording session begun once the mice were introduced in the arena containing the blotting papers.

Video recordings were analyzed with ANY-maze software (Stoelting Europe, Dublin, Ireland), a video tracking system detecting the center of the animal as reference point, and the following parameters were quantified: the total distance travelled, the number of entries in the central zone, the distance travelled in the central zone and the number of entries in the danger zone and the total time of freezing.

Test sessions (H_2O vs. H_2O / H_2O vs. H_2S , 25 or 125 μM) were performed systematically at least 2–3 hours after the control sessions (H_2O vs. H_2O) to avoid the order of cues exposure as a potential confounder. For each behavioral parameter, the data was processed in two steps to allow for comparisons across animals and experimental conditions²³. First, for each mouse, the value obtained during the control session (H_2O vs. H_2O) was set to 100%, and the corresponding value from the test session (either H_2O vs. H_2O or H_2O vs. H_2S) was scaled proportionally. A difference index was then calculated for each individual by subtracting the normalized control value (100%) from the normalized test value, yielding a relative change in percentage. This normalization step accounted for inter-individual variability and differences in units between behavioral parameters. Secondly, we used the formula: $Index_i = (X_{test,i} - X_{control,i}) - \mu_{H_2O}$, where $X_{test,i}$ represents the normalized value obtained for the test, $X_{control,i}$ the normalized value obtained for the control session and μ_{H_2O} represents the mean of the reference H_2O , in order to center the H_2O distribution at 0% and to express the effect of H_2S relative to this baseline. Negative index values for distance-related parameters and number of entries indicate increased fear-related behaviors, such as reduced exploration. Positive index values for the freezing parameter reflect fear responses, as freezing is a well-established indicator of fear. The place of the safety zone was randomly changed between the mice.”